# NL2Repo-Bench: Towards Long-Horizon Repository Generation Evaluation of Coding Agents

**Jingzhe Ding**[1][*]  **Shengda Long**[1][2][*]  **Changxin Pu**[1][*]  **Ge Zhang**[1][3][*]  **Huan Zhou**[1][*]  **Hongwan Gao**[1]  **Xiang Gao**[1]
**Chao He**[1]  **Yue Hou**[3][4]  **Fei Hu**[1]  **Zhaojian Li**[1]  **Weiran Shi**[1]  **Zaiyuan Wang**[5]  **Daoguang Zan**[1]
**Chenchen Zhang**[3]  **Xiaoxu Zhang**[1]  **Qizhi Chen**[2][3]  **Xianfu Cheng**[4]  **Bo Deng**[4]  **Qingshui Gu**[1]  **Kai Hua**[1]
**Juntao Lin**[6]  **Pai Liu**[3]  **Mingchen Li**[6]  **Minghao Li**[6]  **Xuanguang Pan**[4]  **Zifan Peng**[6]  **Yujia Qin**[1]  **Yong Shan**[1]
**Zhewen Tan**[3]  **Haoran Wang**[3]  **Zihan Wang**[6]  **Weihao Xie**[3]  **Yishuo Yuan**[7]  **Jiayu Zhang**[3]  **Yunfei Zhao**[6]
**He Zhu**[3]  **Liya Zhu**[1]  **Chenyang Zou**[6]  **Ming Ding**[1]  **Jianpeng Jiao**[5]  **Jiaheng Liu**[3][7]  **Minghao Liu**[6]  **Qian Liu**[1][3]
**Chongyang Tao**[3][4]  **Jian Yang**[3][4]  **Tong Yang**[2][3]  **Zhaoxiang Zhang**[3]  **Xinjie Chen**[1]  **Wenhao Huang**[1]

## Abstract

Recent advances in coding agents suggest rapid progress toward autonomous software development, yet existing benchmarks primarily evaluate short-horizon behaviors such as localized code generation, scaffolded completion, or repository repair, leaving it unclear whether agents can sustain coherent reasoning, planning, and execution over the extended horizons demanded by real-world repository construction. To address this gap, we introduce **NL2Repo-Bench**, a benchmark explicitly designed to evaluate the long-horizon repository generation from scratch: given only a single natural-language requirements document and an empty workspace, agents must autonomously design the architecture, manage dependencies, and produce a fully installable Python library. Experiments across state-of-the-art open- and closed-source models reveal that long-horizon repository generation remains largely unsolved, with even the strongest agents achieving merely 40% average test pass rates and rarely completing an entire repository correctly. Further analysis identifies systematic long-horizon failure modes, including premature termination, loss of global coherence, fragile cross-file dependencies, and inadequate planning over hundreds of interaction steps. These results position NL2Repo-Bench as

a rigorous, execution-based testbed for evaluating sustained agentic competence and highlight long-horizon reasoning as a key bottleneck for autonomous coding agents. Our data and code are available at https://github.com/multimodal-art-projection/NL2RepoBench.

## 1. Introduction

Large language models (LLMs) have rapidly evolved from passive code completion tools into increasingly autonomous coding agents capable of planning, editing, executing, and validating software (OpenAI, 2025; Anthropic, 2025b; Liu et al., 2025a; Wang et al., 2025; Yang et al., 2024; 2025b; Huang et al., 2025). This progress has shifted the frontier of automated programming from short-horizon, localized tasks toward a more ambitious goal: end-to-end software construction driven solely by natural-language intent. Achieving this goal requires not only strong code synthesis capabilities, but also sustained long-horizon reasoning, global planning, and cross-file consistency—capabilities that are central to the vision of autonomous software engineering and, more broadly, Artificial General Intelligence (AGI).

Despite this progress, the evaluation landscape has not kept pace with the capabilities being claimed. Most existing benchmarks for coding agents emphasize short-horizon behaviors, such as generating individual functions (Chen et al., 2021; Austin et al., 2021), completing partially specified modules (Liu et al., 2025b; 2024), or repairing bugs within pre-existing repositories (Jimenez et al., 2024b; Deng et al., 2025). While valuable, these settings significantly reduce the demands placed on long-term planning and system-level coherence by providing strong structural priors, limited temporal scope, or frequent human intervention. As a result, *they do not adequately measure whether an agent can sustain coherent decision-making over the hundreds of steps*

[*]Equal contribution  [1]Bytedance Seed  [2]Peking University  [3]M-A-P  [4]Beihang University  [5]Humanlaya Data  [6]2077-AI  [7]Nanjing University.  Correspondence to: Xinjie Chen <chenxinjie.bj@bytedance.com>, Wenhao Huang <huang.wenhao@bytedance.com>, Ge Zhang <zhangge.eli@bytedance.com>.

*Proceedings of the 43rd International Conference on Machine Learning*, Seoul, South Korea. PMLR 306, 2026. Copyright 2026 by the author(s).

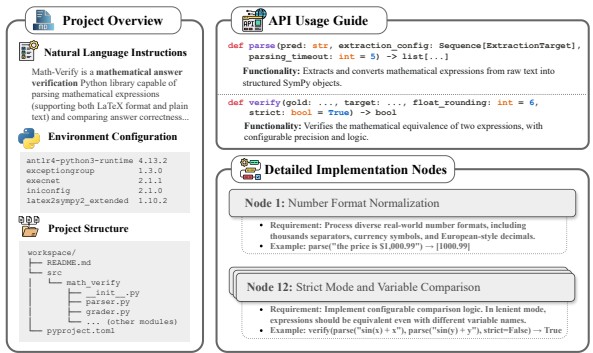

*Figure 1.* An example NL2Repo task document, illustrating the structured specification (project description, supports, and API usage guide) that agents receive before repository generation.

*required to design, implement, debug, and finalize a complete software repository.*

Recent repository-level benchmarks partially address this gap, but important limitations remain. Some rely on scaffolded project structures or predefined function signatures (Zhao et al., 2024), converting the task into constrained code infilling rather than autonomous construction. Others depend on LLM-based evaluators or qualitative judgments (Starace et al., 2025; Patwardhan et al., 2026; Zhang et al., 2025a; Chou et al., 2025; Fu et al., 2025), which introduce bias and obscure true functional correctness. Although recent works have expanded evaluation to visual artifacts (Zhang et al., 2025b) and web agent interactions (Li et al., 2025), rigorous evaluation of full-repository construction from natural language remains underexplored. Even benchmarks that use unit tests often assume the presence of an existing codebase (Du et al., 2023; Jimenez et al., 2024b), shifting the challenge toward repair or regression rather than creation. Consequently, a fundamental question remains unanswered: *can current coding agents reliably generate a complete, installable software repository from scratch while maintaining long-horizon coherence?*

To address this, we introduce **NL2Repo-Bench**, a benchmark designed to evaluate the long-horizon repository generation capabilities of coding agents. In NL2Repo-Bench, an agent is provided with a single natural-language requirements document and an empty workspace. From this minimal starting point, the agent must autonomously perform architectural design, dependency management, multi-file implementation, and packaging to produce a fully functional Python library. Crucially, no project scaffolding, source code, or test cases are revealed during development, forcing the agent to reason globally and persistently across the entire construction process. Besides, evaluation in NL2Repo-Bench is strictly execution-based. Each generated repository is verified against the original upstream pytest suite of a real-world open-source project, executed within a controlled environment. This design ensures an objective and

binary notion of correctness grounded in real software behavior, rather than subjective judgments or proxy metrics. Moreover, the benchmark comprises 104 tasks drawn from diverse application domains and varying complexity levels, with input documents averaging nearly 19k tokens, reflecting the scale and ambiguity of realistic software specifications.

Through extensive experiments on state-of-the-art (SOTA) open- and closed-source models within multiple agent frameworks (AI, 2024; Anthropic, 2024; Wang et al., 2025), we find that long-horizon repository generation remains a major unsolved challenge. Even the strongest agents achieve average test pass rates below 40% and rarely succeed in fully reproducing a repository. Beyond aggregate performance, our analysis reveals systematic long-horizon failure modes, including premature termination due to overconfidence, loss of global architectural consistency, brittle dependency handling, and an inability to persistently execute and verify plans over extended interaction sequences.

By explicitly targeting long-horizon reasoning and execution, NL2Repo-Bench provides a missing evaluation axis for coding agents and complements existing function-level (Chen et al., 2021; Austin et al., 2021) and repair-focused benchmarks (Jimenez et al., 2024b). We argue that progress on NL2Repo-Bench will require advances beyond larger context windows, including improved agentic planning, robust self-verification loops, and mechanisms for maintaining global coherence over long development trajectories. As such, NL2Repo-Bench serves both as a diagnostic tool for current systems and as a guiding benchmark for future research on autonomous, long-horizon software engineering.

To summarize, our contributions are as follows:

- We formalize the NL2Repo task as constructing a software repository from an empty workspace given only a single requirements document, and introduce NL2Repo-Bench, a strictly verifiable and long-horizon agentic coding benchmark, which requires the generation of a complete, installable Python library that passes the upstream `pytest` suite.

- We release a reverse-engineered, quality-assured corpus of tasks and a standardized evaluation image that isolates environment effects, enabling apples-to-apples comparisons across agents.

- We provide baseline results with SOTA coding agents, revealing substantial gaps versus repair/completion settings and highlighting open challenges in architecture, dependency management, and cross-file consistency.

## 2. Related Works

### 2.1. Coding Benchmarks for LLMs

Function-level benchmarks such as HumanEval and MBPP primarily assess localized reasoning for isolated programming tasks, abstracting away repository-level constraints (Chen et al., 2021; Austin et al., 2021). At the repository scale, prior work clusters into three paradigms. (i) **Repair/Regression**: the SWE-bench series (Jimenez et al., 2024a; Zan et al., 2026; Miserendino et al., 2025; Deng et al., 2025; Xu et al., 2025) evaluate resolving real issues within existing projects and validating fixes against project test suites. (ii) **Completion/Auto-completion**: RepoBench (Liu et al., 2024) measures the capacity to complete incomplete projects by generating missing components within real repositories (Liu et al., 2025b). (iii) **Paper-to-repository Reproduction**: PaperBench (Starace et al., 2025) tests whether agents can replicate AI research by constructing a repository and executing experiments based on papers, with performance often judged by LLMs on code and results rather than authoritative upstream tests. In a different direction, Commit0 (Zhao et al., 2024) targets from-scratch library generation but provides project structure and function signatures as strong priors. These settings differ along critical axes of input priors, evaluation signals, and output granularity.

### 2.2. LLM-Driven Coding Agents

The rise of LLMs has given birth to coding agents that assist or autonomously perform software development. Beyond IDE-embedded assistance, more autonomous approaches have emerged. SWE-agent introduces an Agent–Computer Interface (ACI) that abstracts file search, navigation, editing, and testing, enabling LLMs to operate on repository-level tasks and achieve strong results on SWE-bench (Yang et al., 2024). OpenHands further generalizes this paradigm as a framework for end-to-end autonomous development, allowing agents to plan, code, and validate with minimal human intervention (Wang et al., 2025).

## 3. NL2Repo-Bench

To evaluate repository-level coding abilities with verifiable ground truth, NL2Repo-Bench derives tasks from real-world Python libraries characterized by modular architectures and authoritative `pytest` suites, as shown in Figure 2. Agents receive only a single natural-language specification and must reconstruct the complete repository from scratch, including file structures and functional logic. Correctness is strictly measured by executing the generated code against the original upstream tests. In the following sections, we detail the pipeline for task selection, document generation, and quality assurance (Section 3.1), followed by the statistical

characteristics of the resulting dataset (Section 3.2).

### 3.1. Benchmark Construction

**Repository Selection.** To ensure that each task reflects a realistic and sufficiently challenging repository-level generation scenario, we curate a set of Python open-source libraries from GitHub as the targets to be reproduced. Our selection procedure follows four principled criteria designed to guarantee task complexity, stability, and testability. The detailed criteria of selection can be found in Appendix C.1.

After initial selection, human annotators clone each candidate project and perform an preliminary review of its structure, dependency, and overall organization. Once they develop a sufficient understanding of the repository's layout and expected behavior, the annotators run the project's built-in test suite. Only repositories that successfully pass all of their native tests at this stage are deemed qualified and retained as target repositories in our benchmark.

**Project Document Writing.** Upon selecting the target repositories, our annotator construct the input specification through a systematic reverse-engineering process. The goal is to translate the entire repository, including implementation files, core functionality, module relationship, and test logic, into a coherent natural-language (NL) document. This document serves as a high-level functional specification that enables a developer (or agent) to fully reproduce the repository's behavior without accessing the source code. The details of tutorials of our reverse engineering process can be found at Appendix C.3.

To standardize the task formulation, every project document in NL2Repo-Bench is structured into four specific sections:

- **Project Description:** A high-level overview of the repository's goals, scope, and primary functionality.
- **Supports:** Supplementary materials required for development, such as required third-party packages and the expected directory structure.
- **API Usage Guide:** Detailed descriptions of the core features to be implemented, including specific requirements for classes, functions, and their expected behaviors.
- **Implementation Nodes:** Concrete examples or references for critical APIs (provided where applicable) to ground the agent's implementation planning.

Among these sections, the **API Usage Guide** is particularly critical. Since NL2Repo-Bench evaluates generated repositories using the original test suite of the target repository, the guide must accurately cover all functional elements exercised by the tests; any omission would render the task unsolvable. To ensure completeness, we adopt an **AST-assisted** annotation workflow. Annotators first run an automated AST scanner to extract a complete inventory

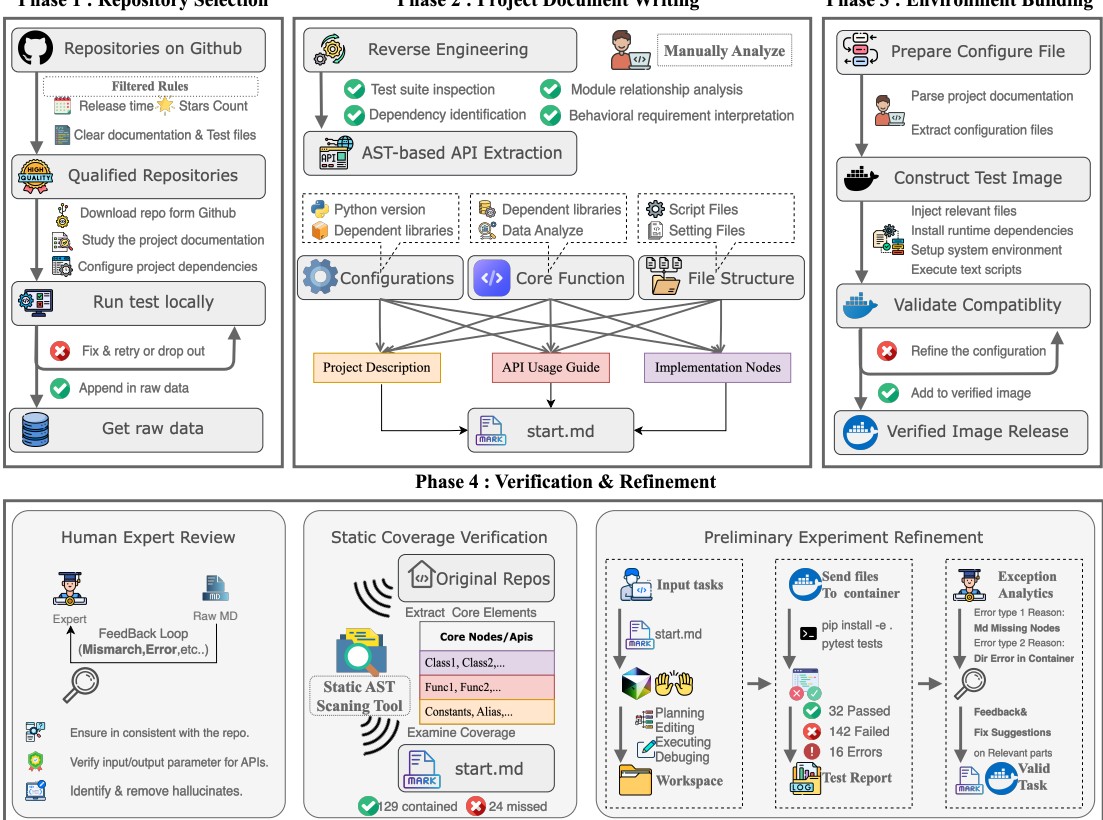

*Figure 2.* Construction pipeline of NL2Repo-Bench.

of classes and functions with their signatures. Using this inventory as a blueprint, annotators then reconstruct each API entry with precise behavioral semantics and I/O specifications. This process guarantees this section is exhaustive and that all nodes strictly align with the original repository, enabling reliable long-horizon repository generation.

**Environment Building.** To ensure deterministic evaluation and strict isolation between code generation and execution environments, we build a dedicated Docker-based test image for each task. The upstream repository is injected into this environment and must pass the official test suite in its entirety. The environment configuration strictly follows the repository's documented dependencies, and any test failures are resolved solely through system-level adjustments (e.g., dependency pinning or library configuration), without modifying the functional source code.

In addition, we implement a strategy of minimizing non-functional build constraints by relaxing or sanitizing rigid checks in build manifests (e.g., mandatory documentation or license files). Such constraints are either removed or satisfied via synthetic artifacts, ensuring that evaluation reflects only the functional correctness of generated code, rather than compliance with extraneous build prerequisites.

**Verification and Refinement.** To ensure the quality and reliability of the constructed tasks, we employ a multi-stage validation pipeline that integrates both human and automated verification. Our quality assurance framework consists of the following steps:

- **Human Expert Review:** Professional Python experts manually verify the fidelity of the source content, the correctness of function signatures(including names, input/output parameters, etc..), and the elimination of hallucinated information, guaranteeing that the document faithfully corresponds to the original codebase.
- **Static Coverage Verification:** To ensure that the specification document is sufficiently comprehensive, we reuse the AST-derived API inventory from the document writing stage as a static verification oracle, comparing all core classes, functions, and method signatures against the specification to ensure they are explicitly documented with correct signatures and semantics. This process formally validates specification completeness before evaluation.
- **Preliminary Experiment Refinement:** We validate each task's feasibility by running SOTA coding agents solely with the provided specification document. The generated workspaces are packaged and executed as libraries within the containerized environments constructed in the pre-

*Table 1.* Task categories and statistics of the NL2Repo-Bench.

| Category | Count |
|---|---|
| Web Development | 10 |
| Testing | 13 |
| Utility Libraries | 11 |
| Machine Learning | 7 |
| Data Analysis & Processing | 18 |
| Database Interaction | 7 |
| Networking Tools | 9 |
| Batch File Processing | 5 |
| System Tools | 24 |
| Overall | 104 |

vious stage, where they are evaluated using the official test suite. Then a senior Python engineer systematically inspects all fails and exceptions. By cross-referencing failure reports with the task documentation and execution environment, the engineer distinguishes issues caused by document-level specification ambiguities or environment misconfigurations, from those attributable to intrinsic model limitations. Based on this diagnosis, we iteratively refine task documents and environments solely to resolve such ambiguities or misconfigurations, ensuring that benchmark failures reflect genuine reasoning or implementation challenges rather than artifacts of task or environment design.

A task is recognized as valid only upon passing all verification stages. Tasks failing any check undergo iterative refinement and re-evaluation until full compliance is achieved.

### 3.2. Benchmark Statistics

Following the rigorous selection, construction, and validation pipeline described in Section 3.1, we assemble the final **NL2Repo-Bench** dataset. The benchmark comprises 104 tasks spanning 9 distinct categories of Python libraries, as detailed in Table 1. NL2Repo-Bench represents the first evaluation framework designed to assess coding agents on their ability to generate fully-functional Python repositories solely from natural-language descriptions. The average input length of a NL2Repo-Bench task is approximately 18,800 tokens, a scale that substantially exceeds the input complexity of existing repository-level benchmarks.

Beyond domain diversity, NL2Repo-Bench covers a wide spectrum of task complexities. We categorize tasks into three difficulty levels(easy, medium, and hard) based on the original project size and total lines of code. Detailed criteria and statistics are provided in Appendix A.

## 4. Experiments

### 4.1. Experimental Settings

In our study, we mainly apply **NL2Repo-Bench** to evaluate the performance of various models within the **OpenHands-CodeAct Agent** framework[1]. The list of models evaluated are listed in Appendix F.

To assess the impact of end-to-end agent frameworks, we additionally evaluate two commercial coding agents, **Cursor-CLI** and **Claude Code**, both using **Claude-Sonnet-4.5** as the underlying language model to control for model capability while isolating the effects of agent architecture and tool integration.

For each task, the agent is launched via a single initial user instruction(specified in Appendix E) in an empty workspace containing only the task specification. The agent then completes the task fully autonomously, without any further human intervention. No restrictions are imposed on tool usage or interaction length. Upon completion, the generated workspace is packaged and evaluated in a dedicated Docker environment using the original upstream `pytest` suite. To improve robustness, we modify the test execution pipeline to ensure that all collected test cases are executed even in the presence of collection errors, preventing isolated failures from dominating the final score. The final performance metric is the average of per-task test pass rates, where each task's score is defined as the fraction of passed test cases out of all tests in the repository.

### 4.2. Main Results

As summarized in Table 2, several noteworthy observations are as follows:

**Coding agents remain far from being able to synthesize full repositories.** As shown in Table 2, all models achieve an average test pass rate below 40.5%, and nearly half of them fall below the 25% threshold. Across the entire set of 104 tasks, the strongest model fully passes the official `pytest` suite for only 5 repositories in a single run (Pass@1). These results indicate that current LLMs and coding agents still lack the robustness, long-horizon planning ability, and cross-file consistency required to generate a complete repository from scratch. Even the best-performing systems struggle to construct end-to-end runnable software purely from natural-language specifications, underscoring the substantial gap that remains in achieving reliable repository-level synthesis.

**Model performance trends.** Across all evaluated systems, the **Claude** series (including Claude-Sonnet-4 and 4.5) consistently achieves the strongest performance. We attribute

---

[1]For model **Gemini-3-pro**, we apply **Cursor-CLI** framework since it suffers from agent-in-a-loop error frequently in Openhands.

*Table 2.* Model performance (Pass Rate %) across difficulty levels. We report the overall pass rate, Pass@1 count, and breakdown by difficulty. Best results are **bolded**. Unless otherwise specified in parentheses, all models use the **OpenHands** agent framework.

| Model | Overall Score (%) | Pass@1 (Count) | Easy ($\leq$1.5k LOC) | Medium (1.5k-4k LOC) | Hard ($\geq$4k LOC) |
|---|---|---|---|---|---|
| Claude-Sonnet-4.5 (Claude Code) | **40.2** | 3 | 51.8 | **44.5** | **25.1** |
| Claude-Sonnet-4.5 | 39.9 | 3 | **55.3** | 43.0 | 21.4 |
| Claude-Sonnet-4.5 (Cursor) | 39.2 | 4 | 52.9 | 41.4 | 24.8 |
| Claude-Sonnet-4 | 37.0 | **5** | 53.1 | 41.3 | 16.1 |
| Gemini-3-pro (Cursor) | 34.2 | 3 | 44.9 | 40.9 | 16.8 |
| DeepSeek-V3.2 | 27.6 | 1 | 43.1 | 29.1 | 12.9 |
| Kimi-k2 | 22.7 | 3 | 40.8 | 20.2 | 11.6 |
| DeepSeek-V3.1 | 22.2 | 1 | 35.7 | 21.6 | 12.1 |
| GPT-5 | 21.7 | 1 | 38.4 | 20.7 | 9.6 |
| Qwen3-Instruct | 17.9 | 1 | 34.7 | 15.2 | 8.9 |
| GLM-4.6 | 17.5 | 2 | 34.4 | 15.5 | 6.5 |
| Qwen3-thinking | 13.8 | 1 | 25.0 | 11.3 | 6.5 |

this advantage to a combination of factors, including robust long-horizon reasoning, stable task execution over extended interactions, and strong alignment between specification understanding and code synthesis. While large context capacity provides useful headroom for handling long specifications and iterative development, our later analyses show that context length alone is insufficient to explain the observed performance gaps (Appendix P). In contrast, **GPT-5** performs notably worse than expected: it frequently halts progress and waits for further input, leading to the second lowest average number of interaction turns (one of only two models below 100 turns; Table 3). Rather than continuing autonomously, GPT-5 often pauses generation when uncertainty arises.

To further assess the stability of model performance, we conduct three independent runs for a subset of representative models. The detailed results are presented in Appendix H.

**Performance degrades with repository complexity.** Evaluating model performance across difficulty tiers reveals a clear monotonic decline as task complexity increases. This trend validates that the NL2Repo-Bench difficulty hierarchy meaningfully reflects the intrinsic challenges of real-world, repository-level development. The pronounced performance drop on harder tasks further underscores current LLM-based coding agents' limitations in long-horizon reasoning, cross-module coordination, and managing dependency-intensive engineering workflows.

**Performance varies substantially across task categories.** Appendix G further breaks down results by library category, revealing that models like Claude-Sonnet-4.5 are particularly strong on system tools and database interaction tasks, while all models struggle on machine learning and networking tasks. This suggests that current agents handle infrastructure-style repositories with clearer packaging and dependency structure better than domains emphasizing complex algorithmic pipelines or protocol-heavy logic.

**Context window size matters.** Context window capacity strongly correlates with NL2Repo-Bench performance, with ultra-long-context models consistently achieving the best results due to their ability to sustain long-horizon repository construction. Nevertheless, it does not determine success alone, as reasoning robustness and agentic persistence remain critical. A detailed analysis is provided in Appendix P.

**Same baseline models exhibit consistent performance across agent frameworks.** As shown in Table 2, Claude-Sonnet-4.5 shows less than 1% performance variance across three agent frameworks, which is negligible given the full-repository construction and multi-test evaluation required by NL2Repo-Bench. In contrast, performance differences between models are substantially larger, indicating that benchmark outcomes are dominated by the intrinsic capabilities of the underlying LLM rather than the choice of agent framework. This suggests that NL2Repo-Bench primarily evaluates base-model reasoning and code generation ability, with agent-level orchestration playing a secondary role.

### 4.3. Analysis and Discussions

#### 4.3.1. TOOLS USED DURING DEVELOPMENT

We analyze the distribution of tool calls across all models within the OpenHands framework, as illustrated in Appendix I. The most frequently used tools during the development process are: `execute_bash`, `str_replace_editor`, and `task_tracker`, which are primarily used for file management & testing, code editing, and planning, respectively.

We analyze the correlation between total tool invocations and model scores shown in Table 9. Among the most frequently used tools, `task_tracker` shows the strongest association(0.711), highlighting the importance of explicit task planning for repository-scale code generation. Although `think` exhibits a high raw correlation, its sparse usage limits the robustness of this signal, suggesting a weaker link to overall performance in this setting.

*Table 3.* Interaction turns statistics and model performance.

| Model | Avg. Turns | Std. Dev. | Max Turns | Score (%) | Turns/ Score |
|---|---|---|---|---|---|
| Claude-Sonnet-4.5 | 181.6 | 64.1 | 394 | 39.9 | 455.1 |
| Claude-Sonnet-4 | 180.7 | 80.3 | 416 | 37.0 | 505.4 |
| Kimi-k2 | 275.1 | 164.6 | 878 | 22.7 | 1211.9 |
| DeepSeek-V3.1 | 202.3 | 170.5 | 990 | 22.2 | 919.6 |
| DeepSeek-V3.2 | 254.3 | 119.9 | 822 | 27.6 | 1145.5 |
| GPT-5 | 78.4 | 29.1 | 165 | 21.7 | 361.2 |
| Qwen3-Instruct | 212.9 | 166.0 | 940 | 17.9 | 1225.4 |
| GLM-4.6 | 138.6 | 110.7 | 533 | 17.5 | 825.0 |
| Qwen3-Thinking | 70.2 | 40.9 | 246 | 13.8 | 508.7 |

To further investigate the impact of the planning tool, we evaluate a subset of representative models with the task tracker disabled. Detailed results are provided in Appendix J.

### 4.3.2. INTERACTION TURNS AND MODEL PERFORMANCE

A critical factor distinguishing model performance on NL2Repo-Bench is the number of interaction turns required to complete repository generation. As shown in Table 3, models exhibit markedly different interaction patterns.

**The Dominance of Premature and Incomplete Task Completion in GPT-5.** A striking finding is GPT-5's significantly lower turn count—averaging only 78.4 turns, which is merely 42% of Claude-Sonnet-4.5's 181.6 turns and the lowest among all evaluated models. Despite this brevity, GPT-5 achieves a moderate score of 0.217, suggesting high per-turn quality but insufficient task completion. Our case analysis reveals that GPT-5 frequently halts prematurely and awaits user confirmation, with 13.4% of tasks exhibiting early termination behavior. This pattern indicates that GPT-5's design prioritizes human-in-the-loop collaboration over fully autonomous task completion, misaligning with NL2Repo's requirement for end-to-end repository generation without human intervention.

**Efficiency vs. Persistence Trade-off.** Apart from GPT-5, Claude-Sonnet-4.5 achieves the best balance between efficiency and performance (turns/score ratio of 455.1), completing tasks with moderate turn counts while maintaining the highest test pass rate. In contrast, DeepSeek-V3.2 employs more turns (254.3) but with mixed results—it achieves the best score in all open-source models, but with a high cost. This suggests that simply increasing interaction attempts does not guarantee success; the quality of planning and execution strategy matters more.

### 4.3.3. THE EARLY TERMINATION AND NON-FINISH PROBLEM

Failure to complete the repository structure is a primary cause of low pass rates. We categorize the reason for incomplete tasks into two distinct behaviors: **Early Termination** and **Non-Finish**, as clarified in detail in Appendix M.

As shown in Figure 7 quantifying these phenomena, following observations emerges:

**Early Termination and Non-Finish as Divergent Failure Modes. Qwen3-Thinking** exhibits both a high Early Termination rate (49.0%) and a comparably high Non-Finish rate (46.2%), revealing a paradox of reasoning-intensive models. Its explicit internal "thinking" process appears to induce premature confidence: rather than validating implementations through execution or testing, the model reasons itself into believing that the task has been completed. We hypothesize a form of *hallucinated verification*, where internal reasoning substitutes for external validation, leading to incomplete repositories despite perceived success. In contrast, **GPT-5** demonstrates the opposite pattern, with a low Early Termination rate (13.4%) but an *overwhelming Non-Finish rate (84.5%)*. Instead of falsely concluding completion, GPT-5 frequently halts progress to solicit user guidance, reflecting a strong *human-in-the-loop* alignment that persists under *"Force-Autonomous"* instructions. While effective in collaborative settings, this conservative strategy is detrimental in our fully autonomous benchmark, where no feedback is provided. Together, these results highlight a fundamental trade-off between overconfident autonomous completion in thinking models and excessive reliance on human intervention in assistive models.

**Persistent Models.** Claude-Sonnet-4.5 reveals the most robust behavior with 1.9% Non-Finish rate and no Early Termination . It consistently drives the development process to a conclusion, correlating with its superior performance.

### 4.3.4. TOOL USAGE PATTERNS AND EFFICIENCY

We analyze tool invocation patterns to understand how different models approach repository generation. Table 8 and Figure 3 present tool usage statistics.

**Code Editing vs. Execution Ratio.** Most models allocate the majority of tool calls to code editing (str_replace_editor), ranging from 48-62% of total calls. The edit-to-execution ratios reveal different development strategies: Claude-Sonnet-4.5 (1.67:1) and GPT-5 (1.62:1) maintain similar ratios, while DeepSeek-V3.2 uses a lower ratio (1.51:1), suggesting more frequent testing and validation cycles.

**Task Planning and Organization.** A notable differentiator is task_tracker usage, which reflects systematic planning capabilities. GPT-5 devotes 14.5% of calls to task track-

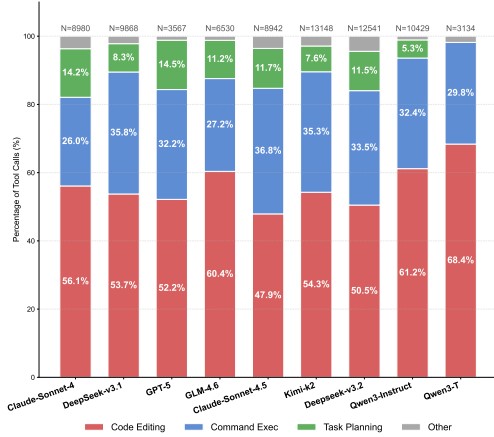

*Figure 3.* Tool usage distribution across models. Most models allocate 48-62% of calls to code editing, 26-37% to command execution, but differ significantly in task planning usage.

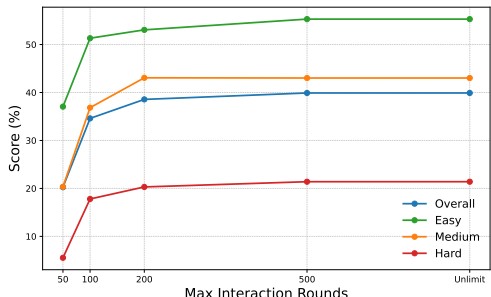

*Figure 4.* The effect of limitations on iteration rounds on the performance of Claude-Sonnet-4.5 on different difficulty level tasks.

### 4.4. Ablation Studies

#### 4.4.1. IMPACT OF INTERACTION ROUND LIMITS

The main experiments permit unlimited interaction rounds, allowing agents to iteratively refine the repository without budget constraints. To assess whether high scores depend on extensive trial-and-error rather than genuine reasoning, we re-evaluate each model under varying maximum round limits. This ablation quantifies the degree to which agents rely on long iterative loops and examines their effectiveness under more realistic, bounded interaction settings.

As shown in Figure 4, model performance increases steadily across all difficulty levels as the interaction limit is raised from 50 to 200 rounds. Once the limit reaches 200 rounds—which is slightly above Claude-Sonnet-4.5's average number of interactions under the unrestricted setting—further expanding the maximum round budget yields only marginal improvements. These findings indicate that while interaction budget matters in the low-round regime, model performance quickly saturates once the budget exceeds its natural working range, suggesting that the primary limitations lie in semantic reasoning, architectural planning, and cross-file consistency rather than in the sheer number of allowed interaction steps.

#### 4.4.2. IMPACT OF REVEALING ALL TEST CASES.

By default, agents only receive the natural-language specification, while the pytest suite remains hidden to mimic real development settings. To estimate an upper bound on performance and determine how much models struggle with implicit requirement inference, we run an additional condition where all test cases are made visible during generation.

Results in Table 10 show that exposing the full test suite yields substantial performance gains: the pass rate of Claude-Sonnet-4.5 (in Claude Code framework) increases markedly from 40.2% to 59.4%, and the number of fully-passed tasks jumps from 3 to 18. This trend is consistent with the expected benefit of providing test cases, which can

ing—the highest among all models—followed by Claude-Sonnet-4 (14.2%). In stark contrast, Qwen3-Thinking allocates 0 to planning, relying instead on its internal reasoning mechanism. This disparity correlates with performance: models with higher planning tool usage (Claude, GPT-5) achieve better scores, while those neglecting explicit planning (Qwen3-Thinking) suffer from premature termination and incomplete implementations.

**Quantity vs. Quality Trade-off.** Figure 6 plots total tool calls against final scores, revealing a non-monotonic relationship. GPT-5 achieves a moderate score (0.217) with fewer calls (3567), demonstrating high per-call quality. Kimi-k2 makes the most calls (13148), but with divergent outcomes, indicating that the sheer quantity of attempts cannot compensate for poor strategy. Claude-Sonnet-4.5's 8942 calls yield the best score, establishing an efficiency baseline.

**Agentic Workflow Patterns.** Beyond aggregate tool usage, the *sequence* of tool call actions reveals the agent's underlying reasoning strategy. We analyze the transition probabilities between consecutive tool calls to identify distinct workflow patterns, as detailed in Appendix N.

#### 4.3.5. FAILURE TAXONOMY

To understand *why* agents fail, we categorized the error types encountered during the evaluation of generated repositories. Beyond function-level benchmarks where `AssertionError` dominates, NL2Repo-Bench reveals a more complex failure landscape. Detailed typical failure patterns and behaviors can be found at Appendix Q.

guide and constrain both the development process and the resulting implementation. However, it is equally noteworthy that—even under this "cheating" scenario where evaluation-phase tests are made available during development—the model's overall score still does not exceed 60%. This indicates that, despite the advantages of test-driven development, generating a fully functional, end-to-end runnable repository from scratch remains a substantial challenge for current coding agents, pointing to fundamental limitations in long-horizon coordination and large-scale code synthesis rather than merely missing supervision signals.

## 5. Conclusion

We introduce **NL2Repo-Bench**, a benchmark for evaluating whether LLMs and agents can autonomously generate complete, installable Python repositories from a single natural-language specification, with correctness strictly verified by upstream test suites. Experimental results reveal a substantial gap between current SOTA and the demands of long-horizon, repository-level generation, even at the highest observed pass rates. Our analysis identifies two dominant failure modes: premature termination driven by overconfidence in internal reasoning, and failures of fully autonomous execution stemming from implicit human-in-the-loop assumptions. These findings suggest that progress on NL2Repo-Bench requires advances beyond scale alone, including stronger agentic planning, self-correction, and environment management. We release our benchmark to support future research in end-to-end autonomous software engineering.

## Impact Statement

Our work contributes to the study of autonomous software engineering by introducing a rigorous benchmark for repository-level code generation with verifiable ground truth. By emphasizing end-to-end repository construction, long-horizon planning, and evaluation via authoritative upstream test suites, NL2Repo-Bench provides a standardized and reproducible platform for analyzing the strengths and limitations of current LLMs and coding agents. We believe this benchmark will facilitate more reliable empirical comparisons, encourage the development of models with stronger planning and consistency capabilities, and support future research on building more robust, transparent, and trustworthy AI systems for software development.

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

## A. Difficulty Level in NL2Repo Tasks

We classify the difficulty of tasks in NL2Repo based on the number of lines of code (LOC) in the original repository. As shown in Table 4, repositories with fewer than 1,500 LOC are categorized as **Easy**, while those exceeding 4,000 LOC are categorized as **Hard**. Repositories falling between these thresholds are considered **Medium**. The number of tasks for each difficulty level is shown at Table 4.

*Table 4.* Difficulty statistics of the NL2Repo benchmark.

| Difficulty Level | LOC Range | #Tasks |
|---|---|---|
| Easy | $\leq$ 1500 LOC | 26 |
| Medium | 1500–4000 LOC | 46 |
| Hard | $\geq$ 4000 LOC | 32 |

## B. Available Tools in OpenHands-CodeAct Framework

In our experiments, agents interact with the environment using the following standardized tools provided by the OpenHands-CodeAct framework:

- **execute_bash**: Executes a bash command in a persistent shell session. This tool is essential for file navigation, package installation, and running test suites.
- **think**: Enables the agent to articulate internal reasoning traces and plan next steps without executing any changes in the environment.
- **finish**: Signals the completion of the current task.
- **browser**: Allows the agent to interact with a web browser using Python code (e.g., for documentation lookup).
- **execute_ipython_cell**: Runs a cell of Python code in an interactive IPython environment.
- **task_tracker**: Provides structured task management capabilities, allowing the agent to view, add, and update the status of development tasks.
- **str_replace_editor**: A custom file editing tool designed for viewing, creating, and editing files. It uses strict string matching to ensure precise code modifications.
- **fetch**: Retrieves content from a specified URL and optionally extracts it as markdown.
- **create_pr / create_mr**: Tools to simulate the submission of a Pull Request or Merge Request on platforms like GitHub, GitLab, or Bitbucket.

## C. Tutorial for NL2Repo-Bench Annotators

This section provides the guidelines and step-by-step workflow used by annotators to construct task specifications for NL2Repo-Bench. The goal is to ensure that all specifications are (1) semantically faithful to the original repository, (2) sufficiently comprehensive for end-to-end repository development, and (3) consistent across annotators. The process consists of three major phases: project selection, repository comprehension and environment validation, and structured specification writing. The detailed tutorial is shown below.

### C.1. Phase 1: Project Selection

Following the criteria in Section 3.1 of the main paper, annotators first determine whether a candidate GitHub repository is eligible for inclusion. The selected repository must satisfy all the following criteria:

- **Complexity.** Each repository must contain 300 to 120,000 lines of code. This range excludes trivial projects while avoiding extremely large systems that exceed the context windows of current coding agents, ensuring tasks remain both meaningful and tractable.
- **Maturity.** We require each repository to have at least 10 GitHub stars, which serves as a minimal proxy for community adoption, maintenance quality, and functional reliability.
- **Completeness and Testability.** A repository must include `pytest`-based test cases, and its official version must successfully pass all of them. This requirement ensures that the behavioral ground truth is both well-defined and verifiable.
- **Recency.** Only repositories that have been created or updated within the past three years are considered, allowing

NL2Repo-Bench to reflect contemporary coding practices and avoid outdated dependency ecosystems.

This includes checking project maturity, testing completeness, license compatibility, and the feasibility of isolating the core functionality.

### C.2. Phase 2: Repository Understanding & Test Validation

Annotators must obtain a precise and executable understanding of the target repository before writing its specification.

**(1) Local Setup and Preliminary Analysis.**

- Download the repository locally from GitHub.
- Conduct an initial review of the project, including its purpose, core functionality, directory structure, and external dependencies.

**(2) Environment Construction.** Annotators create an isolated environment based on the repository's documentation (e.g., `README`, `requirements.txt`, `setup.py`, or `pyproject.toml`). All relevant dependencies, including possible undocumented ones, must be installed.

**(3) Full Test Execution.** Annotators must run *all* existing test cases in the repository. A repository is considered valid only if all tests pass. If failures occur, annotators must diagnose and resolve environment-related issues (e.g., Python version mismatches, incompatible dependency versions, missing system packages). Repositories with unresolvable failures are excluded.

### C.3. Phase 3: Task Specification Construction

Each task requires a comprehensive specification consisting of four parts: a project-level description, support information, API-level usage guide, and implementation nodes.

#### C.3.1. PROJECT DESCRIPTION

Annotators provide a high-level functional summary of the repository, describing:

- The overall purpose and design of the project;
- Its major components and interaction patterns.

Annotators may reference official documentation or use LLM-based tools to draft an initial summary, but the final text must be manually verified to ensure factual correctness.

#### C.3.2. SUPPORT INFORMATION

**(1) Third-Party Dependencies.** Annotators list all external libraries needed to run the project, including explicit dependencies and any additional packages required for the tests to pass. Version numbers must be preserved where applicable.

**(2) Repository File Structure.** A complete directory layout of the target codebase is included (excluding the testing directory). All files relevant to the implementation must be listed to ensure that the model can reconstruct the same structure during development.

#### C.3.3. API USAGE GUIDE

This component provides natural-language documentation for all functional units in the repository.

**(1) Static Extraction of Functional Nodes.** Annotators run a static analysis tool to extract:

- All classes, functions, and constants;
- Their names and signatures;
- Their file locations.

Test directories are excluded from the scan.

**(2) Node-Level Documentation.** For each functional node, annotators write a standalone description containing:

- The name and purpose of the node;
- Input arguments and return values (with additional explanation if complex);
- A functional description aligned strictly with the implementation.

Text may be drafted with AI tools for assistance, but annotators must manually verify that every detail matches the underlying source code exactly.

**(3) Module Import Instructions.** Annotators provide examples of how modules and APIs should be imported. These import paths are derived from the project's internal import patterns as observed in the test files (excluding imports from external libraries).

### C.3.4. IMPLEMENTATION NODES

Annotators provide concrete examples or references for critical APIs (provided where applicable) within the target repository to help understand the specific usage of APIs. When completing this part, annotators could refer to the examples in the repository.

### C.4. Quality Requirements

All descriptions must be:

- Fully consistent with the repository's implementation;
- Complete with respect to all functional nodes;
- Free of speculative or missing behaviors not grounded in the source code.

This ensures that the resulting specification is both comprehensive and faithful, enabling an agent to implement the entire repository solely from the provided instructions.

## D. Annotator Team and Background

NL2Repo-Bench is annotated by a team of more than **50** contributors. The majority are **graduate students** (Master's or PhD) in computer science or related fields, with a small number of senior undergraduate students serving as annotators under the same qualification standards. All annotators possess at least three years of Python programming experience and familiarity with real-world software engineering practices, ensuring the reliability and technical accuracy of the annotations. Annotators are compensated fairly in accordance with their workload.

## E. User Instructions for NL2Repo-Bench Tasks

> **User Input**
>
> According to the start.md in the workspace, implement the entire project as per the requirements specified in the document, ensuring that the final product can be directly run in the current directory. The running requirements should comply with the <API Usage Guide>section of the document. Please complete this task step by step.

## F. Models Evaluated on NL2Repo-Bench

To ensure the comprehensiveness of our experimental results, the evaluation encompassed the following models:

*Open-source models* — DeepSeek-V3.1, DeepSeek-V3.2(Liu et al., 2025a), Qwen3-235B-Instruct[2],Qwen3-235B-Thinking[3] (Yang et al., 2025a), Kimi-k2 (Team et al., 2025b), and GLM-4.6 (Team et al., 2025a).

*Closed-source models* — Claude-Sonnet-4, Claude-Sonnet-4.5 (Anthropic, 2025a;b), Gemini-3-pro (Google Cloud, 2025) and GPT-5 (OpenAI, 2025).

---

[2]abbr. Qwen3-Instruct
[3]abbr. Qwen3-thinking or Qwen3-T

*Table 5.* Model performance (Pass Rate %) across different task categories.

| Model | Web Dev | Testing | Utility Libs | ML | Data Proc. |
|---|---|---|---|---|---|
| Claude-Sonnet-4.5 (Claude Code) | 56.9% | 30.4% | 59.8% | 19.7% | 36.7% |
| Claude-Sonnet-4.5 | 37.3% | 31.9% | 52.4% | 16.2% | 36.8% |
| Claude-Sonnet-4.5 (Cursor) | 44.0% | 33.0% | 53.8% | 9.4% | 34.6% |
| Claude-Sonnet-4 | 48.2% | 27.5% | 51.2% | 8.5% | 41.3% |
| Gemini-3-pro (Cursor) | 33.8% | 28.2% | 43.9% | 10.0% | 26.0% |
| DeepSeek-V3.2 | 33.6% | 21.7% | 44.4% | 11.1% | 20.8% |
| Kimi-k2 | 29.5% | 18.7% | 27.3% | 7.5% | 24.8% |
| DeepSeek-V3.1 | 21.2% | 16.0% | 30.8% | 10.5% | 18.2% |
| GPT-5 | 26.0% | 18.6% | 28.0% | 7.0% | 31.6% |
| Qwen3-Instruct | 18.2% | 15.2% | 18.7% | 8.0% | 18.9% |
| GLM-4.6 | 17.7% | 14.4% | 29.5% | 8.4% | 14.1% |
| Qwen3-thinking | 15.5% | 9.1% | 19.2% | 7.4% | 11.6% |

| Model | DB Interact. | Network | Batch Ops | Sys Tools | - |
|---|---|---|---|---|---|
| Claude-Sonnet-4.5 (Claude Code) | 41.3% | 23.4% | 43.3% | 43.6% | - |
| Claude-Sonnet-4.5 | 44.4% | 30.5% | 43.7% | 50.3% | - |
| Claude-Sonnet-4.5 (Cursor) | 46.9% | 25.4% | 35.0% | 49.7% | - |
| Claude-Sonnet-4 | 33.1% | 23.4% | 32.0% | 43.2% | - |
| Gemini-3-pro (Cursor) | 40.8% | 28.3% | 44.0% | 45.2% | - |
| DeepSeek-V3.2 | 22.5% | 16.9% | 36.4% | 34.1% | - |
| Kimi-k2 | 23.8% | 15.6% | 37.0% | 22.3% | - |
| DeepSeek-V3.1 | 25.0% | 14.8% | 33.7% | 28.0% | - |
| GPT-5 | 23.4% | 10.4% | 18.9% | 20.2% | - |
| Qwen3-Instruct | 24.4% | 14.2% | 32.6% | 17.6% | - |
| GLM-4.6 | 28.9% | 0.6% | 20.5% | 21.0% | - |
| Qwen3-thinking | 16.6% | 6.4% | 16.3% | 15.8% | - |

*Table 6.* Performance stability across repeated runs on NL2Repo-Bench.

| Model | Single Run | 3-Run Mean | Std. Dev. |
|---|---|---|---|
| DeepSeek-V3.2 | 27.6 | 27.3 | 3.7 |
| Kimi-k2 | 22.7 | 23.3 | 4.2 |
| Qwen3-T | 13.8 | 13.7 | 3.5 |

## G. Performance of Models across different task categories

The performance of models across 9 different categories is shown at table 5.

## H. Results of 3-Trial Experiments on Selected Models

We select three representative models and conduct experiments with three independent runs. The results are reported in Table 6. We can see that the average performance across 3 runs is consistently close to the corresponding single-run result for all evaluated models. Moreover, the observed standard deviations are relatively small, suggesting that the benchmark exhibits limited variance across repeated runs and that single-run evaluations provide a reasonably reliable estimate of model performance.

## I. Distributions of Tools used in NL2Repo-Bench tasks within Openhands-CodeAct framework.

The times of all the tools in used by different models in all NL2Repo-Bench tasks is shown in Figure 5, while the detailed statistics of the usage of key tools in the tasks is shown in Table 8.

## J. Impact of Explicit Planning via Task Tracker

To further investigate whether explicit planning contributes to performance, we conduct an experiment by disabling the `task_tracker` tool while keeping all other settings unchanged in Openhands-Codeact framework. We evaluate the two

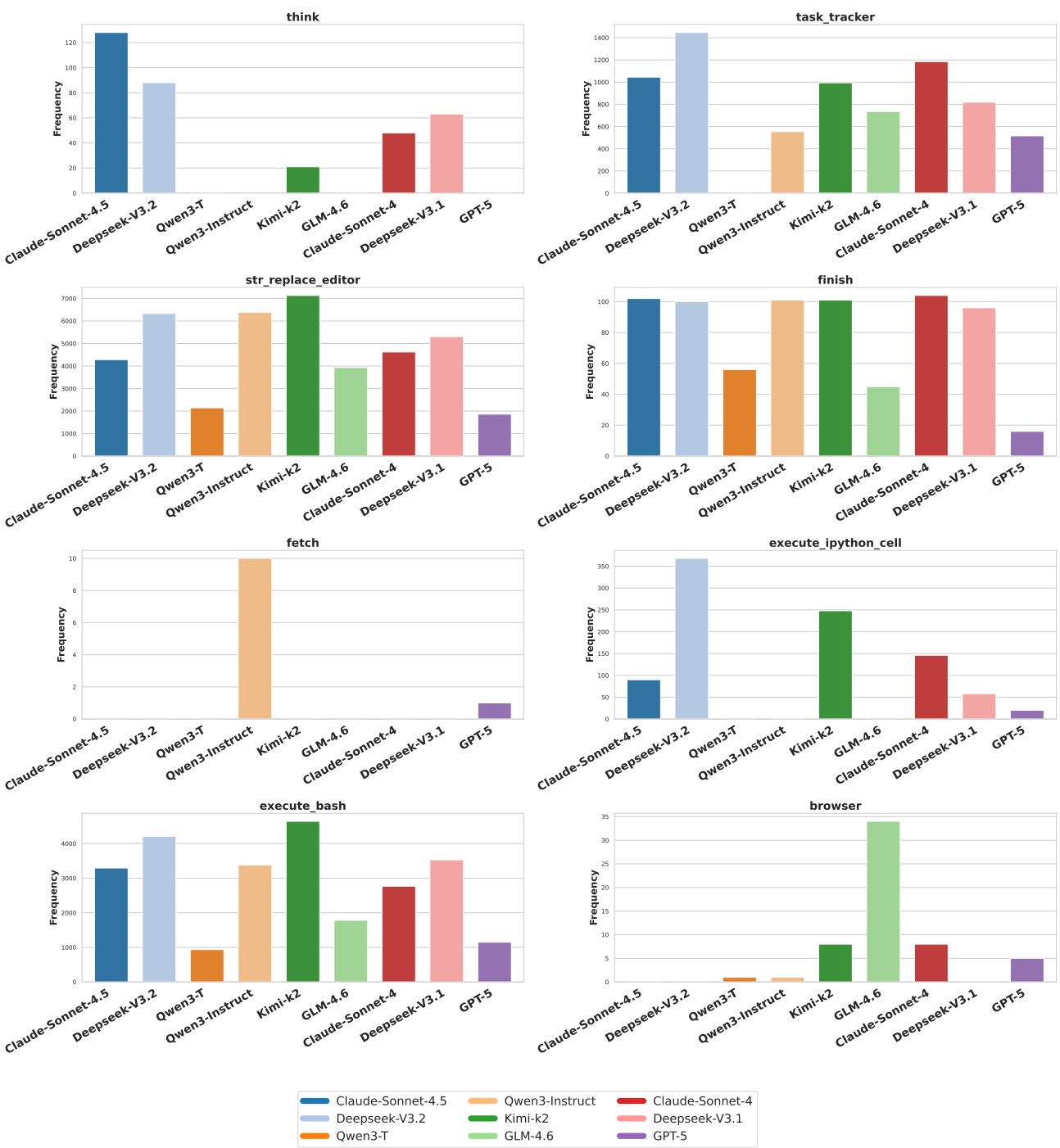

*Figure 5.* The total times using different tools for models in all NL2Repo-Bench Tasks.

models that frequently use this tool.

As shown in Table 7, removing `task_tracker` consistently reduces performance across both models. Although agents may still perform implicit planning without the tool, the observed degradation suggests that explicit plan decomposition and progress tracking provide measurable benefits for repository-level code generation.

*Table 7.* Performance impact of disabling `task_tracker`. Scores are reported as overall benchmark performance (%).

| Model | With Task Tracker | Without Task Tracker |
|---|---|---|
| GPT-5 | 21.7 | 18.7 |
| Claude-Sonnet-4 | 37.0 | 34.0 |

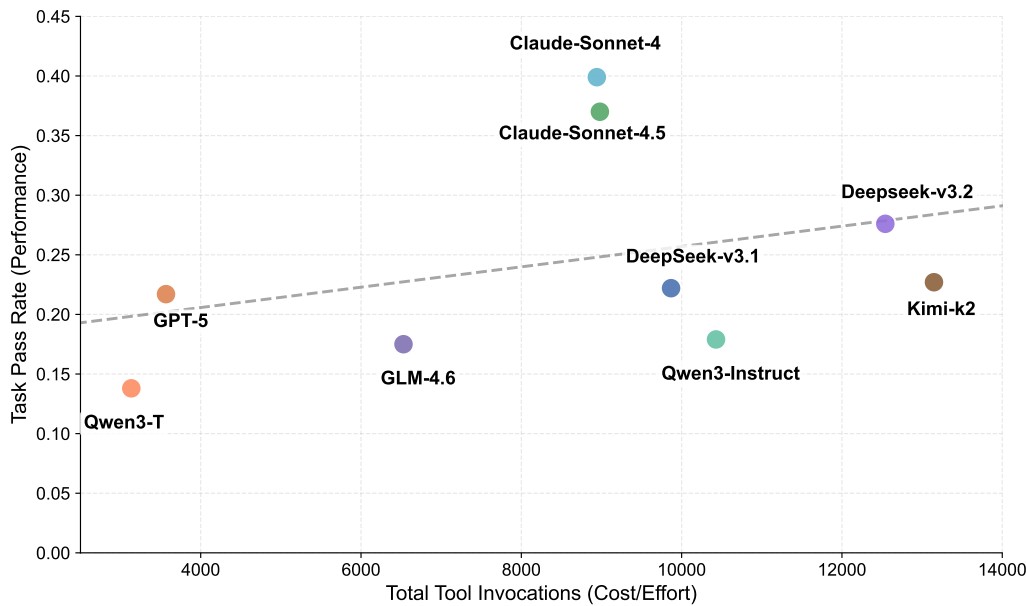

*Figure 6.* Tool call efficiency: quantity vs. quality. The dashed line represents all model's average performance.

## K. Visualization of Tool Call Efficiency of All Models

The relationship between socres and tool call times for all models in OpenHands-CodeAct framework is shown at Figure 6. GPT-5 and Claude-series operates above this line, while most other models fall below it.

*Table 8.* Tool usage statistics across models (104 tasks).

| Tool | Claude-Sonnet-4 | DeepSeek-v3.1 | GPT-5 | GLM-4.6 | Deepseek-V3.2 | Qwen3-Instruct | Qwen3-T | Kimi-k2 | Claude-Sonnet-4.5 |
|---|---|---|---|---|---|---|---|---|---|
| str_replace_editor | 4623 | 5304 | 1861 | 3955 | 6333 | 6379 | 2143 | 7134 | 4284 |
| execute_bash | 2766 | 3528 | 1148 | 1781 | 4205 | 3382 | 934 | 4642 | 3293 |
| task_tracker | 1185 | 819 | 516 | 735 | 1447 | 554 | 0 | 994 | 1045 |
| Other | 306 | 217 | 42 | 79 | 556 | 114 | 57 | 378 | 320 |
| **Total** | 8880 | 9868 | 3567 | 6530 | 12541 | 10429 | 3134 | 13148 | 8942 |
| **Avg/Task** | 85.38 | 94.88 | 34.30 | 62.79 | 120.59 | 100.28 | 30.13 | 126.42 | 85.98 |

## L. Correlation between different Tool Call Frequency and Model Scores

The correlation between toolcall times and model performances is shown at Table 9.

## M. Early Termination and Non-Finish Phenomena

- **Early Termination (Overconfidence):** The agent explicitly invokes the `finish` action, claiming the task is done, but does so prematurely (defined here as fewer than 100 interaction turns). This typically indicates a "false positive" estimation of progress.
- **Non-Finish (Passive Failure):** The agent never invokes the `finish` tool. Instead, the session ends because the agent

*Table 9.* Correlation between tool invocation frequency and model performance.

| Tool | Correlation | #Models Not Using This Tool |
|---|---|---|
| think | 0.816 | 4 |
| task_tracker | 0.711 | 1 |
| finish | 0.512 | 0 |
| execute_bash | 0.371 | 0 |
| browser | -0.264 | 3 |
| execute_ipython_cell | 0.402 | 3 |
| fetch | -0.1917 | 7 |
| str_replace_editor | 0.161 | 0 |

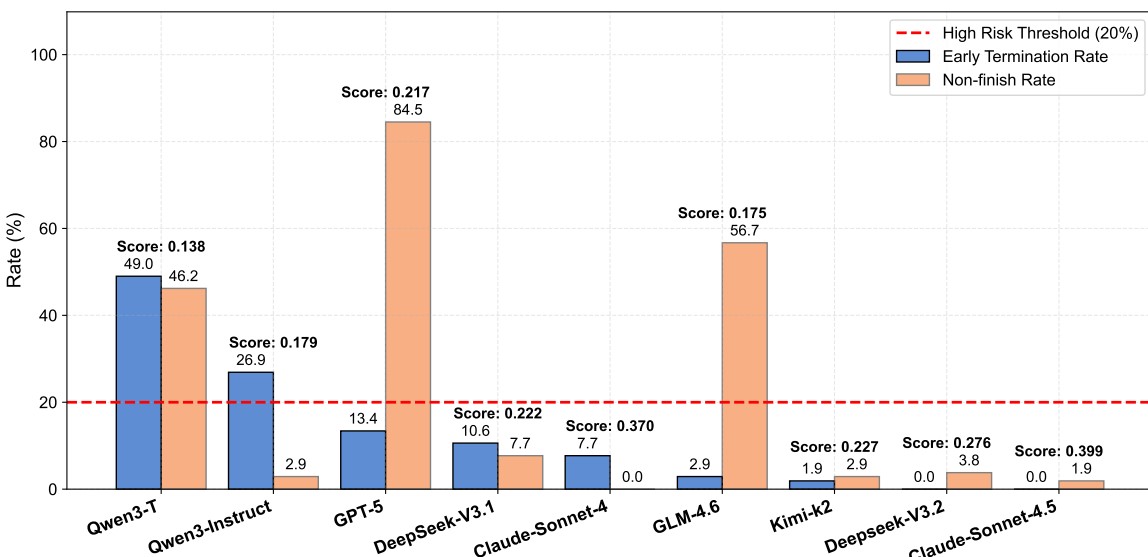

*Figure 7.* Early termination and non-finish rates across models. Thinking models (e.g., Qwen3-T) exhibit an alarmingly high early termination rate of 49.0% and non-finish rate of 46.2%, leading to unsatisfactory scores.

halts to await user input (e.g., asking for clarification or confirmation) or reaches a system timeout. This behavior reflects a lack of agency or an inability to proceed autonomously.

## N. Agentic Workflow Patterns

Beyond aggregate tool usage, the *sequence* of actions reveals the agent's underlying reasoning strategy. We analyze the transition probabilities between consecutive tool calls to identify distinct workflow patterns.

**The "Edit-Test" Loop.** High-performing models like Claude-Sonnet-4.5 exhibit a strong cyclic pattern between str_replace_editor and execute_bash (specifically pytest). This "Edit-Test" loop indicates a Test-Driven Development (TDD) or rapid feedback strategy, where the agent verifies changes immediately after implementation.

**The "Navigation" Trap.** In contrast, lower-performing models show high transition probabilities between execute_bash (ls/cd) and read_file, often without intervening edit actions. This "Navigation Loop" suggests the agent is struggling to locate relevant files or build a mental map of the repository, leading to wasted context and interaction turns.

**Blind Editing.** We also observed a "Blind Editing" pattern in some models, characterized by consecutive str_replace_editor calls without intermediate testing. This often leads to accumulated errors that are difficult to debug later in the session.

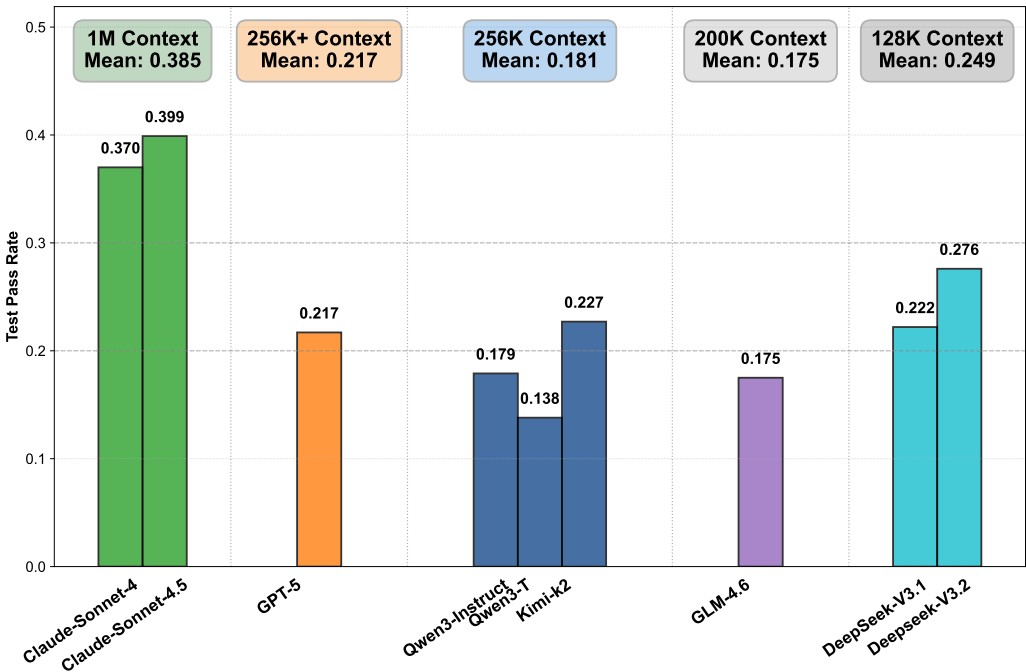

*Figure 8.* Impact of context window size on performance. The 1M context model (Claude) substantially outperforms 256K models. GPT-5 (256K+) underperforms despite a larger context, suggesting context size is necessary but not sufficient.

## O. Behavior of Model Performance when All Test Cases Available

The score of Claude-Sonnet-4.5 (Claude-Code framework) on NL2Repo-Bench when revealing all test cases is shown in Table 10. The Average Score and Pass@1 both increase accordingly.

*Table 10.* Comparison of the performance of Claude-Sonnet-4.5(Claude Code) on whether revealing all test cases.

| Inputs | Easy | | Medium | | Hard | | Overall | |
|---|---|---|---|---|---|---|---|---|
| | Avg Score (%) | Pass@1 | Avg Score (%) | Pass@1 | Avg Score (%) | Pass@1 | Avg Score (%) | Pass@1 |
| Document only | 51.8 | 1 | 44.5 | 1 | 25.1 | 1 | 40.2 | 3 |
| Document + unittest | 73.2 | 9 | 67.5 | 7 | 35.6 | 2 | 59.4 | 18 |

## P. Impact of Context Window Size

Context window capacity emerges as a crucial factor in NL2Repo-Bench performance. Models can be broadly categorized by their context capabilities: those with ultra-long context ($\geq$ 1M tokens, e.g., Claude series, Gemini-3-pro) and those with standard context (e.g., DeepSeek, GPT-5, Qwen). Figure 8 visualizes performance differences.

**The Long-Context Advantage.** The top tier of the leaderboard is exclusively occupied by long-context models. Claude-Sonnet-4.5 (40.2% on Claude Code) and Gemini-3-pro (34.2%) substantially outperform the standard-context cohort. This advantage stems from NL2Repo's demanding context requirements: average task documents contain 18,800 tokens, generated code can span 10,000-100,000 tokens, and over 150 average interaction turns accumulate approximately 100000 tokens of conversation history. While 128K capacity theoretically suffices for snapshots, the 1M+ window provides critical headroom for maintaining full context throughout extended development sessions, enabling better cross-file consistency and architectural coherence.

**Context Size Is Necessary But Not Sufficient.** However, a large context window does not guarantee superior performance. For instance, despite supporting longer contexts, Kimi-k2 achieves a pass rate of only 22.7%, lagging behind DeepSeek-V3.2 (27.6%) which operates within a 128K limit. Similarly, GPT-5 (21.7%) underperforms compared to the top-tier models and even open-source baselines. This suggests that while context capacity provides the *potential* for maintaining global

coherence, the model's underlying reasoning capability and agentic behavior (e.g., persistence vs. early termination) are equally critical determinants of success.

**Context Amplifies Planning Effectiveness.** As shown in Figure 3, while explicit planning (via `task_tracker`) is a strategy employed by several models regardless of their context size—GLM-4.6 (11.2%), GPT-5 (14.5%), and Claude-Sonnet-4.5 (11.7%) all exhibit high usage—the outcomes differ significantly. Standard-context models like GLM utilize planning tools frequently but achieve lower success rates (17.5%), suggesting that without a massive context window to retain the full history of the plan and its execution states, the effectiveness of planning diminishes over long horizons. In contrast, Claude's 1M+ window allows it to maintain a persistent and coherent view of the task lifecycle, maximizing the utility of its planning actions.

## Q. Typical Failure Patterns of Agent-generated Workspaces

**Environment and Dependency Issues.** A significant portion of failures stem from `ImportError` or `ModuleNotFound` exceptions. This highlights a key challenge in repository-level generation: agents often struggle to correctly structure the package (e.g., missing `__init__.py`) or manage internal dependencies between modules, resulting in code that is logically correct but structurally broken.

**Test Suite Alignment.** Another common failure mode is the mismatch between the agent's implementation and the official test suite's expectations (e.g., function signatures or class attributes). This suggests that while agents can follow the natural language instructions, they may miss subtle constraints required by the pre-existing tests.

