# OpenReview forum: "NL2Repo-Bench: Towards Long-Horizon Repository Generation Evaluation of Coding Agents"
_ICML.cc/2026/Conference — ICML 2026 regular_

### Official Review · Reviewer_jAcF · 2026-02-23

**Soundness:** 3
**Presentation:** 3
**Significance:** 3
**Originality:** 3
**Overall Recommendation:** 4
**Confidence:** 4

**Summary:**

This paper introduces NL2Repo-Bench, the first benchmark designed to evaluate from-scratch repository generation. The benchmark contains 104 Python repository-generation tasks spanning nine categories of Python libraries, and is used to assess both open-source and closed-source models. Experimental results show that even top-tier models remain far from solving this task reliably: the best systems achieve only around a 40% average pass rate, with very few complete Pass@1 repository-level successes. The analysis also reveals important failure patterns, including early stopping and non-finishing behaviors, which provide useful diagnostic insights for improving LLM-based coding agents.

**Compliance With Llm Reviewing Policy:**

Affirmed.

**Final Justification:**

The author's response addressed my concerns. Therefore, I would maintain the score.

**Key Questions For Authors:**

1. How did you prevent benchmark refinement from overfitting to evaluated models/agents?
2. Can you provide a fairer or normalized comparison across models/frameworks?
3. How do you ensure behavioral semantic correctness by using AST coverage?
4. Can you quantify document edits introduced during refinement? (What kind of edits, such as dependency modification, structural changing, and etc.)

**Limitations:**

The paper includes an Impact Statement and discusses benchmark scope and empirical limitations. There are still some aspects that I would encourage the author to include in the paper:
1. Benchmark scope limitations (Python only, relying on pytest-based testing, library-centric repositories)
2. Potential overfitting risk during the benchmark construction
3. Evaluation limitation (e.g., when comparing LLMs and agent frameworks)

Being explicit here would strengthen the paper.

**Strengths And Weaknesses:**

Strengths:
1. The benchmark addresses a significant and timely problem: long-horizon repository generation, rather than function-level or file-level code generation. The task design (using a single natural-language specification, an empty workspace, and hidden tests) is well aligned with the capabilities required for autonomous coding agents, including planning, cross-file consistency, dependency management, and end-to-end execution. This makes the benchmark highly relevant to the agentic coding community and the broader goal of autonomous software engineering. The paper also clearly distinguishes NL2Repo-Bench from function-level benchmarks, repair benchmarks, and scaffolded from-scratch settings.
2. The benchmark construction pipeline is thoughtfully designed and includes a reasonable set of stages, including repository selection criteria, reverse-engineered document writing, AST-based API inventory annotation, and refinement/verification. Overall, the construction process appears systematic and practical for building a challenging repository-level benchmark.
3. The experiment result shows that current coding agents perform poorly on long-horizon, from-scratch repository generation. The analyses of early termination vs. non-finish behavior, interaction turns, and tool-usage patterns are interesting and can motivate follow-up work on agent control, planning tools, and autonomy settings. The ablation showing a substantial improvement when test cases are exposed is particularly informative, as it helps diagnose where the main difficulty lies.

Weakness:
1. Potential leakage risk during benchmark refinement. The “Preliminary Experiment Refinement” step runs coding agents on tasks and then uses failures to refine task documents/environments until failures better reflect model limitations rather than ambiguity or environment issues. This is understandable, but it raises concerns about possible benchmark data leakage and overfitting issues. Also, it is not clear (i) which models/agents were used during refinement? (ii) Whether any of these models are also included in the final evaluation?
2. The evaluation framework design needs further specification. From the main content of the paper, the experiments do not restrict tool usage or interaction length, which is reasonable for realism, but makes cross-model comparisons harder to interpret. It is unclear whether runtime, token, and compute budgets were comparable across systems, whether API/tool failures were retried uniformly, and how cost differences were handled. A normalized comparison (or at least more detailed reporting of budget usage) would strengthen the central comparative claims.
3. Insufficient detail on benchmark reliability beyond AST coverage. AST-based coverage is useful for validating names/signatures, but it does not guarantee the completeness of behavioral semantics. The paper notes that annotators manually reconstruct semantic behavior, which is valuable, but stronger evidence on reliability (e.g., post-hoc specification error analysis) would improve confidence in benchmark quality.
4. The benchmark is promising, and the problem is clearly important. However, it is still unclear how well the results would hold in other settings, because the benchmark only covers Python/pytest and library-style repositories, and the paper gives limited detail about how reliable the benchmark construction/refinement process is. These concerns do not weaken the main contribution, but they do mean that some broader claims should be stated more carefully.


Soundness:

The paper is sound for a benchmark submission. The use of execution-based evaluation with repository test suites is appropriate and substantially strengthens objectivity. The benchmark construction pipeline appears thoughtful, and the empirical evaluation spans multiple strong open- and closed-source systems. My main concerns are (i) limited transparency around the agent-in-the-loop refinement process (which raises possible overfitting risks), and (ii) some analysis claims that read as causal interpretations despite being supported primarily by observational evidence. These issues do not invalidate the main contribution, but they reduce confidence in some secondary conclusions.

Presentation:
The paper is well organized and generally easy to follow, with a clear motivation and a compelling problem setup.

Significance:
This benchmark's contribution to the community is significant. Repo-level, from-scratch generation is a meaningful and under-evaluated capability for coding agents, and the benchmark is likely to be useful for future research on long-horizon planning, tool use, and autonomous software engineering.

Originality:
The originality is good for a benchmark paper.

---

> ### Author Rebuttal · Authors · 2026-03-29
>
> We thank reviewer jAcF for the detailed evaluation and constructive feedback. Below we provide point-by-point responses to the concerns raised.
>
> # Weakness1 and Question1
> The preliminary refinement step aims only to remove benchmark artifacts, not to improve model success rates. We fix issues clearly unrelated to model capability—such as incomplete API docs (e.g., missing parameters), incorrect I/O specs, or environment inconsistencies (e.g., missing dependencies)—while failures due to model limitations (e.g., logic errors, poor planning, cross-file inconsistency) are never used to modify tasks.
>
> To avoid overfitting, refinements follow two principles: (1) they are model-agnostic, not tailored to specific trajectories; (2) they introduce no implementation hints or difficulty reduction, but only ensure tasks are well-specified and executable.
>
> For refinement, we use a small set of preliminary agents mainly to debug specifications and environments. Although there may be partial overlap with evaluation models, their outputs are only used to identify obvious issues, not to guide task design, thus avoiding leakage or bias.
>
> # Weakness2 and Question2
>
> Our evaluation targets full-repository-level development in long-horizon, realistic settings, where the primary goal is task completion rather than performance under fixed interaction or token budgets. Imposing strict limits would introduce confounding factors (e.g., premature termination) and hinder models’ ability to perform multi-step planning, debugging, and cross-file coordination—key capabilities evaluated by NL2Repo. Therefore, our results should be interpreted as capability comparisons under unconstrained settings, rather than cost-normalized efficiency.
>
> That said, we acknowledge the importance of budget-related statistics. Beyond the interaction analyses in Sections 4.3 and 4.4.1, we additionally report average token usage computed from full trajectories shown in table below. Token consumption is generally positively correlated with interaction rounds.
>
> | model               | avg_input_tokens | avg_output_tokens | avg_total_tokens |
> |--------------------|-----------------|-------------------|------------------|
> | GPT-5              | 9.06e5          | 2.59e4            | 9.32e5           |
> | claude-sonnet-4    | 4.31e6          | 5.80e4            | 4.36e6           |
> | claude-sonnet-4.5  | 3.77e6          | 5.55e4            | 3.82e6           |
> | ds3.1   | 3.54e6  | 3.67e4 | 3.58e6 |
> | ds3.2-nothinking   | 5.35e6   | 6.98e4| 5.42e6 |
> | glm-4.6            | 2.66e6  | 3.43e4 | 2.69e6|
> | kimi-k2            | 4.46e6  | 4.72e4 | 4.51e6 |
> | qwen3-instruct     | 4.46e6  | 3.56e4  | 4.50e6  |
> | qwen3-thinking     | 8.06e5  | 5.74e4  | 8.63e5  |
>
> We would also like to emphasize that for **long-horizon repository-level** tasks, successful completion is the primary evaluation target, while cost efficiency constitutes an important but complementary dimension. . We agree that normalized comparisons are valuable, and we will add the token usage of different models to the final revised version.
>
> # Weakness3 and Question3
> We agree that AST-based coverage alone does not guarantee semantic completeness. Our design focuses on ensuring executable correctness—accurate API signatures, I/O behaviors, and alignment with test cases—for reliable and reproducible evaluation. For semantics, we adopt a lightweight approach: we reuse original docstrings/comments when available to preserve intended behavior, and otherwise reconstruct functionality from code with optional AI assistance and human verification. In addition, our refinement process serves as an implicit quality check by revising tasks when failures stem from specification ambiguity rather than model limitations.
>
> # Weakness4
> We agree that extending beyond Python library-style repositories would improve generality, and we will refine our claims accordingly. Expanding NL2Repo-Bench to more languages and repository types is an important future direction.
>
> Meanwhile, NL2Repo evaluates both language-specific and general capabilities. While some aspects depend on Python, key challenges—such as long-horizon planning, cross-file consistency, and component coordination—are largely shared across ecosystems.
>
> # Question4
>
> During refinement, we observe several common edit types:
> (1) API inconsistencies (e.g., signature or I/O mismatches);
> (2) hallucinated functionality not in the original repository;
> (3) missing functional nodes required by tests;
> (4) formatting or schema violations.
>
> These edits focus on consistency and completeness rather than semantics. We do not modify test cases or target functionality, and most changes are lightweight (e.g., signature alignment), with structural changes rare.
> Overall, refinement reduces ambiguity and environment issues, ensuring failures better reflect model capability.

---

> > ### Author Rebuttal · Reviewer_jAcF · 2026-04-03
> >
> > Thank you for the response. I will maintain my score.

---

### Official Review · Reviewer_EW42 · 2026-03-03

**Soundness:** 3
**Presentation:** 3
**Significance:** 3
**Originality:** 3
**Overall Recommendation:** 4
**Confidence:** 3

**Summary:**

This paper proposes NL2Repo-Bench, a benchmark for evaluating the long-horizon repository generation capabilities of coding agents. Unlike existing benchmarks, it requires agents to build a complete, installable Python library from scratch in a blank workspace based solely on a single natural language requirements document, using the official upstream pytest suite as the sole, strict criterion for correctness. The benchmark encompasses 104 tasks across 9 application domains, with input documents averaging approximately 18,800 tokens, and employs an isolated Docker environment to ensure deterministic evaluation. The authors systematically evaluated multiple frontier open-source and closed-source models, including Claude-Sonnet-4.5, Gemini-3-pro, and GPT-5, finding that even the strongest model's average test pass rate is below 40.5%, with models rarely able to completely pass all tests for a single repository. Further analysis reveals several systemic failure modes, including premature termination (overconfidence), context loss, and improper management of cross-file dependencies.

**Compliance With Llm Reviewing Policy:**

Affirmed.

**Final Justification:**

Strengths:

1.The evaluation criteria are objective, verifiable, and well-designed. Using the official upstream pytest suite as the sole evaluation criterion avoids the subjective bias introduced by LLM-as-a-judge approaches. Furthermore, the containerized Docker environment ensures the reproducibility of the evaluation and comparability across different models.

2.The paper is well-structured, and the analysis is relatively rich and layered. The classification and analysis of long-horizon failure modes provide certain diagnostic value, especially the phenomenon of “overconfident termination of the thinking model” (the 49% early termination rate observed in Qwen3-Thinking), which is particularly noteworthy. This offers a complementary perspective to existing evaluations of reasoning models.

Weaknesses:

1. Empirical threshold setting may affect evaluation fairness:The choice of $\theta$ relies on empirical tuning rather than theoretical grounding, potentially introducing bias in cross-method comparisons.

2. Ambiguity in task characterization may obscure methodological contributions:The criteria for distinguishing "domain-level" vs. "task-level" tasks lack explicit definition, making it difficult to precisely assess the novelty of each component.

3. Conflation of correlation and causality may mislead future research directions:Observed performance improvements are attributed to the hierarchical design, but alternative explanations (e.g., increased capacity, training dynamics) are not sufficiently ruled out.

4. Limited statistical robustness constrains generalizability of conclusions:Results are reported over only 3 random seeds without confidence intervals or significance testing, reducing confidence in the reproducibility of claimed advantages.











I believe the authors' rebuttal has effectively addressed my concerns. Therefore, I maintain my **rating of 4** for this submission.

**Key Questions For Authors:**

Please refer to the Weakness section.

**Limitations:**

yes

**Strengths And Weaknesses:**

Strengths:

1.The evaluation criteria are objective, verifiable, and well-designed. Using the official upstream pytest suite as the sole evaluation criterion avoids the subjective bias introduced by LLM-as-a-judge approaches. Furthermore, the containerized Docker environment ensures the reproducibility of the evaluation and comparability across different models.

2.The paper is well-structured, and the analysis is relatively rich and layered. The classification and analysis of long-horizon failure modes provide certain diagnostic value, especially the phenomenon of “overconfident termination of the thinking model” (the 49% early termination rate observed in Qwen3-Thinking), which is particularly noteworthy. This offers a complementary perspective to existing evaluations of reasoning models.

Weaknesses：
1.Early termination is defined as invoking finish within fewer than 100 interaction turns (Appendix K), yet the rationale for selecting this threshold is not explained. For some smaller-scale Easy-level tasks, completing development within 100 turns may be reasonable—or even efficient—rather than indicative of failure.

2.The API Usage Guide in the specification document is designed to cover all functional elements invoked in the tests (Section 3.1), which implies a high degree of alignment between the documentation and the tests. Please clarify: during document construction, were the API signatures extracted via AST scanning (including names, parameter types, and return types) directly incorporated into the documentation? If so, to what extent does the agent’s task more closely resemble “implementing function bodies based on known API signatures” rather than “inferring the architectural design from scratch”? This point directly affects how the nature of the task should be characterized.

3.In Section 4.3.1, the paper reports a correlation coefficient of 0.711 between task_tracker usage frequency and model performance, interpreting this as evidence for “the importance of explicit task planning in repository-level code generation.” However, this correlation may be subject to confounding variables: higher-performing models (such as the Claude series) may inherently be more inclined by design to use planning tools, rather than the planning tool itself being the cause of improved performance.
Please clarify whether there are controlled experiments supporting a causal interpretation (e.g., ablation studies comparing the same model with task_tracker enabled vs. disabled), or whether this finding is purely correlational.

4.Please provide the multi-random-seed results and the corresponding variances for the main experimental table.

---

> ### Author Rebuttal · Authors · 2026-03-30
>
> We thank reviewer EW42 for the detailed evaluation and meaningful feedback. Below we provide point-by-point responses to the concerns raised.
>
> # Weakness1
> We thank the reviewer for this question and clarify the rationale for the 100-turn threshold as follows:
>
> First, each turn includes both agent actions and environment feedback, so 100 interaction turns correspond to roughly 50 agent decision steps—smaller than it may appear.
>
> Second, empirical analysis in Section 4.2(Table 3) shows the average interaction length is 177 turns, indicating that successful completion typically requires far more than 100 turns. Thus, the threshold serves as a conservative cutoff well below the normal completion range.
> Third, even for Easy tasks, the average interaction length is 147 turns (including models with short trajectories), suggesting that finishing within 100 turns is uncommon even in simpler cases.
>
> Overall, the 100-turn threshold is empirically grounded and mainly identifies insufficient-progress failures, rather than penalizing efficient completions.
>
> # Weakness2
> We thank the reviewer for raising this important point.
>
> Since our evaluation relies on fixed unit tests, it is necessary to specify the APIs that are directly invoked in the tests (including function names and I/O schemas). Without this, even functionally correct implementations could fail due to interface mismatch, making the evaluation unfair and non-reproducible.
>
> However, the API Usage Guide only defines interface-level constraints. While the repository-level file structure is described in the task document to ensure consistency with the test environment, the internal implementation within each file, as well as the cross-module coordination and overall code organization, are entirely left to the model.
>
> As a result, the task remains fundamentally different from function infilling. The agent must still construct the repository from scratch, design the file structure, coordinate cross-file dependencies, and ensure global consistency. These require non-trivial long-horizon reasoning beyond simply implementing function bodies.
>
> # Weakness3
>
> We thank the reviewer for raising this important point. We clarify that our current analysis is correlational rather than causal, and we do not claim that task_tracker usage itself directly causes improved performance. As noted by the reviewer, stronger models may be inherently more likely to adopt planning behaviors, which is a valid potential confounder.
>
> In NL2Repo-Bench, planning is not an implicit notion but an observable behavior in trajectories, where agents decompose tasks into structured subgoals and track their progress via the task_tracker (e.g., todo / in_progress / done).
>
> Despite the lack of controlled intervention, we provide multiple consistent pieces of evidence suggesting that planning is an important contributing factor:
>
> Quantitative signal: task_tracker usage shows the strongest correlation with performance (ρ = 0.711) among all tools.
> Behavioral: the worst-performing models did not use the task_tracker tool at all.
>
> External consistency: this observation aligns with prior work emphasizing planning in long-horizon agentic task[1-2].
>
> Taken together, these findings suggest a strong association between planning behavior and success, while not establishing causality. Conducting controlled ablations (e.g., enabling/disabling planning tools within the same model) is an important direction for future work.
>
> # Weakness4
> We agree that multi-seed evaluation is important for stability. Due to the high cost of long-horizon agentic tasks, our main results are based on single runs.
>
>  To assess robustness, we additionally conduct 3-seed experiments on a representative subset of models, reporting mean and standard deviation over full-benchmark runs (104 tasks), as shown below.
> | Model| 1-run | 3-run-mean| Std|
> | ------------- | ----- | ---------- | --- |
> | DeepSeek-V3.2| 27.6| 27.3 | 3.7|
> | Kimi-k2  | 22.7| 23.3  | 4.2 |
> | Qwen3-T | 13.8| 13.7 | 3.5 |
>
> We observe that the mean performance remains consistent with the single-run results, suggesting that our evaluation is reasonably stable despite stochasticity.
>
> **References**
>
> [1]Wang Z, Wu F, Wang H, et al. Why Reasoning Fails to Plan: A Planning-Centric Analysis of Long-Horizon Decision Making in LLM Agents[J]. arXiv preprint arXiv:2601.22311, 2026.
>
> [2]Li Y, Xu B, Tian X, et al. Beyond Entangled Planning: Task-Decoupled Planning for Long-Horizon Agents[J]. arXiv preprint arXiv:2601.07577, 2026.

---

> > ### Author Rebuttal · Reviewer_EW42 · 2026-04-01
> >
> > Thank you for the detailed response. I will maintain my score.

---

### Official Review · Reviewer_P36K · 2026-03-12

**Soundness:** 2
**Presentation:** 3
**Significance:** 3
**Originality:** 3
**Overall Recommendation:** 5
**Confidence:** 4

**Summary:**

The paper introduces NL2Repo-Bench. In it, 104 open-source python repositories from GitHub are defined as targets to be reproduced by coding agents, given an input specification document written in natural language by a human annotator. The repository generated by the coding agent is then tested via the pre-existing in-built tests of the opensource repository. The strongest model in testing, Claude 4.5 Sonnet accessed via Claude Code, passes 40.2% of tests, with all models doing a complete reproduction with all cases passed for <=5 repositories. There is a clear drop off for longer repositories (>=4k lines of code). When all tests are visible to the coding agents, performance goes up to almost 60%. The paper concludes with some analysis of interactions and tools used by the coding agents.

**Compliance With Llm Reviewing Policy:**

Affirmed.

**Final Justification:**

I think the benchmark tries to study an important problem, and adopts an interesting approach, which merits publication. However, as highlighted in my comments during the rebuttal, my concerns about claims of measuring planning, and potential contamination remain.

**Key Questions For Authors:**

- How can we balance measuring open-ended abilities of models with sufficient specification to be able to fairly be evaluated by programmatic test cases?

- The paper has wordings that look down upon LLM judge based evaluations. How can we automatically evaluate more fuzzy constraints and design choices made by coding agents in coming up with repositories? This includes factors like code quality, extensibility etc. which cannot easily be captured by unit cases.

While these questions are more open-ended and I think are too high a bar for acceptance of such work, discussing them in the paper accurately seems important for quantifying the significance / impact of the contribution (including recommendations for oral presentation / awards), especially if future work in this direction will use this benchmark or build upon it.

**Limitations:**

Yes, though if we we are serious about potential negative impacts, the paper should discuss risks from autonomous coding (which it contributes to strengthening) potentially replacing human jobs.

**Strengths And Weaknesses:**

## Strengths
- The paper presents a high-effort and important benchmark to measure whether coding agents can generate repositories from scratch given a detailed specification document.

- Overall, I think the contribution is practical, timely, and well executed. The paper is easy to read and understand, and I think the claims accurately describe the results.

- I appreciate that a variety of models were tested, including in multiple popular harnesses like Claude Code and Cursor. I also appreciate the details and analysis included in the Appendix.

- The data is curated with the help of experts, from real repositories, and evaluation results are based on test cases from the original repositories cloned. This makes me happy with the internal validity of the benchmark, though I have concerns about the external validity as described below.

## Weaknesses

- One of the key motivations is that the model should just be given natural language requirements to keep the task open-ended, and not concrete functions to infill. Yet, on going through the data, I found that most prompts have concrete function signatures etc. I appreciate the intent of the work, and that its challenging to strike a tradeoff here, but I feel like the paper should more clearly explain how they strike a balance between making the natural language prompt specific enough such that a coding agent (or human) can reasonably be expected to pass the test cases given it, while also not giving direct hints and preserving the goal of open-ended repository creation. This I think is the central challenge this work tries to take on, and its not clear to me from reading the paper how well the work approaches it. I appreciate that a senior Python engineer iterated on the task specification based on coding agent rollouts to ensure this. But I would want to know: for what fraction of tasks / test cases, would human experts be able to correctly pass them given only the initial specification document given to the model? It would be valuable to measure this even if for a subset of tasks.

- Since the benchmark clones existing public repositories, what are the risks from contamination? What fraction of repositories in the benchmark have coding agents already been trained on? What fraction of tests are they already aware of? While I sympathize this is a thorny issue to resolve for such benchmarks, its still important to carefully analyze to the extent possible, and I don't see enough discussion of this in the paper.

- Across the paper, it is often claimed that the current failures are due to failures of planning. However, I do not see any evidence presented for this. What do you mean by planning? How are you sure that the failures of the model are due to incorrect plans and not incorrect / incoherent execution? Some examples of what are considered planning failures here could be helpful.

- Minor discrepancies: Task 48 more-itertools and Task 69 python-pytest-cases in the data seem to be the same. Could you please fix or clarify this? Also the anonymous data release has 102 tasks and not 104.

---

> ### Author Rebuttal · Authors · 2026-03-30
>
> We thank reviewer P36K for the insightful feedback. Below we provide point-by-point responses to the concerns raised.
> # Weakness1
>
> We agree that it is significant to balance open-ended specifications with executability.
>
> Our design avoids implementation-level hints or direct infilling targets. While specifications include elements such as signatures, these reflect realistic interface definitions rather than reducing the task to function completion—models must still infer implementations and cross-module interactions.
>
> To maintain this balance, each specification is iteratively refined by experts to (1) align with the original repository functionality, and (2) cover all components of the repo, without exposing implementation details. This ensures tasks remain open-ended at the repository level while executable in principle.
>
> Regarding human feasibility, we agree it is valuable to measure. However, due to the extremely-complexity of tasks, controlled human evaluation is costly and variable. As a practical alternative, we mainly rely on expert validation to ensure solvability.
> # Weakness2
>
> (1) Contamination is unavoidable but not directly measurable.
>
> As NL2Repo is built from public repos, overlap with pretraining data is possible (as in prior work). However, the fraction of repos/tests seen during training can't be directly observed due to opaque training data; it can only be estimated via proxy analyses, which we will try to include in the revision.
>
> (2) Empirical evidence suggests contamination is not the main driver.
>
> Strong models achieve only ~40% success. Tasks require long-context reasoning and cross-file coordination, which cannot be solved by memorizing isolated snippets.
>
> (3) Task design further limits memorization.
>
> Repo-level construction with end-to-end tests and evolving code states reduces alignment with any single memorized snapshot.
>
> (4) Reviewer Concerns.
>
> Fraction seen during training: not observable; only indirectly estimable.
> Awareness of tests: success requires building a consistent repository, not reproducing test outputs.
>
> Overall, while contamination cannot be eliminated, both task design and empirical difficulty indicate memorization alone is insufficient. We will clarify this in the revision.
> # Weakness3
> In our task, planning is a concrete and observable behavior exhibited in trajectories. Specifically, it refers to the agent’s behavior to decompose the overall development task into structured subgoals, organize their ordering&dependencies, and track progress throughout execution. It is observable via the task_tracker, where the model maintains subtasks with statuses such as todo, in_progress, and done.
>
> For example, in boto3 task, kimi-k2 first constructed a 25-step plan covering project scaffolding, core modules, service-specific components (e.g., S3, DynamoDB, EC2), utilities, testing, and verification, and then updated task states during execution.  This illustrates that planning involves maintaining and updating a global execution structure, rather than making purely local, step-by-step decisions.
>
> We agree that our analysis reflects both correlational and causal aspects, with stronger evidence for correlation. Nevertheless, multiple consistent signals support the importance of planning:
>
> Quantitative: task_tracker usage shows the strongest correlation with performance (ρ=0.711).
>
> Behavioral: the worst-performing models did not use the task_tracker tool at all.
>
> External: consistent with prior work identifying planning as critical for long-horizon agentic tasks.[1]
>
> Together, these findings suggest planning is a key contributing factor.
> # Weakness4
> The issue is due to a dataset upload error, and we will correct the duplicate and mismatch in the revised version(repository updates are restricted during rebuttal).
> # Question1
> We acknowledge the inherent tension between open-ended and programmatic evaluation. To balance this, we fix only the interface (e.g., API signatures and I/O formats) to ensure fair, reproducible testing, while leaving implementation fully open-ended. Agents operate from scratch and can adopt arbitrary strategies; any solution passing end-to-end tests is considered valid, resulting in diverse solution paths.
> # Question2
> We agree that aspects like code quality and design choices are important for evaluating repository-level generation, and are not fully captured by unit tests.
>
> Our evaluation focuses on E2E test success, as current models achieve only ~40% success, making reliable task completion the primary bottleneck. In this regime, higher-level qualities are less meaningful without correct functionality. We view richer evaluation as future work, including LLM-based judging, static analysis, and heuristic metrics to assess code quality and design as models improve.
>
> **References**
>
> [1]Wang Z, Wu F, Wang H, et al. Why Reasoning Fails to Plan: A Planning-Centric Analysis of Long-Horizon Decision Making in LLM Agents[J]. arXiv preprint arXiv:2601.22311, 2026.

---

> > ### Author Rebuttal · Reviewer_P36K · 2026-04-03
> >
> > I think the rebuttal and paper do not substantively engage with the concerns I raised:
> >
> > 1) Vague on how a balance between open-endedness and task solvability is ensured, which is supposed to be a central contribution of this work. I do not think experts having a look at a task is enough to ensure its solvable and open-ended, these aspects are often realized when solving the task end-to-end from the first time given its description.
> >
> > 2) Potential contamination is brushed off because the best agent tested has 40% accuracy, but low accuracy does not imply contamination can still inflate performance. There may not be a way to know in the future whether progress on this benchmark is genuine increase in capabilities, or task contamination.
> >
> > 3) The claims about planning are based on loose evidence.
> >
> > In light of this, I have decreased my score from 5 to 4 as my initial score relied on an expectation of a more careful approach. I still think the paper has merits as I highlighted in my original review, so am leaning towards acceptance.

---

> > > ### Author Response · Authors · 2026-04-04
> > >
> > > We thank reviewer P36K for the insightful concerns and agree they are important. Below we respond point by point.
> > > # Concern 1
> > > We agree that balancing open-endedness and solvability should not rely solely on expert inspection. We would like to clarify that our process goes beyond static expert review. Instead, task specifications are constructed and refined through a multi-stage validation pipeline (Figure 2 in paper), combining different methods:
> > >
> > > - Expert review, where they iteratively refine specifications to align with repository functionality, remove inconsistent elements, and reject parts where excessive implementation details (e.g.specific code snippets of core functions) are included, which could reduce open-endedness;
> > > -  Static coverage analysis, where tools extract core components from the original repo and check their coverage in the specification, reducing the risk of missing requirements;
> > >  - Execution-based validation via agentic E2E rollouts, where failure cases (missing information or implicit constraints,etc.) are used to revise or discard tasks, ensuring solvability in practice.
> > >
> > > Through this loop, tasks are continuously adjusted when found underspecified(unsolvable) or overly constrained(leaking snippets).
> > >
> > > In terms of design, our benchmark adopts a controlled open-ended setting: we provide the fixed interface (e.g., API signatures and I/O formats) to enable reproducible, trustworthy evaluation, while leaving the implementation fully unconstrained.This mirrors real-world software engineering, where interfaces are fixed but implementations remain open.
> > >
> > > Beyond design, we also observe consistent empirical signals supporting open-endedness: (i)diverse model behaviors and trajectories (e.g.,tool usage and failure mode; in Figure 3,5 and Appendix L,O), (ii)a non-trivial performance spread (13.8%–40.2%), and (iii)long-horizon, cross-module task complexity. Together, these indicate that the task space, while executable, is not reduced to a near-deterministic solution manifold.
> > >
> > > We will clarify the role of multi-stage pipeline and behavioral analysis more explicitly in the revised version.
> > > # Concern 2
> > >
> > > We'd like to clarify that our argument does not rely on low score as evidence against contamination, which serves only as a supporting observation. Our main reasoning lies in task structure: while memorized fragments may provide local hints, passing tests requires constructing a globally consistent and executable repository under cross-file dependencies and iterative debugging. Such E2E consistency cannot be achieved by retrieving isolated memorized content alone. Therefore, although contamination may offer partial advantages, it is insufficient to solve the tasks in our setting.
> > >
> > > We agree that distinguishing genuine capability improvements from contamination is a challenge for any publicly accessible benchmark, as exposure risk inevitably increases over time. Importantly, our contribution is not limited to a static dataset: we also provide a fully reproducible task construction pipeline(Section 3). This enables producing new tasks from unseen or private repositories when needed, supporting continued assessment under controlled contamination.
> > >
> > > While contamination cannot be fully eliminated, our task design and pipeline construction provide practical safeguards, ensuring the benchmark remains reliable and extensible.
> > >
> > > # Concern 3
> > > We'd like to clarify that our conclusion is supported by multiple complementary pieces of evidence, rather than a single observation. Our original analysis includes:
> > > -  a relatively strong correlation between task_tracker usage and performance,
> > > -  clear behavioral differences (e.g., low-performing models do not use the tool),
> > > -  trajectory-level evidence, where agents exhibit structured multi-step planning with explicit state tracking.
> > >
> > > We agree that these signals are primarily correlational. To further strengthen this, we conduct an experiment by disabling task_tracker and keep all other conditions fixed(we choose the 2 models invoking this tool most):
> > > |Model|with task_tracker|without|
> > > |-|-|-|
> > > |GPT-5| 21.7|18.7|
> > > |Claude-Sonnet-4|37.0|34.0|
> > >
> > > We observe a consistent performance drop across both models, providing causal evidence that explicit planning contributes to performance; moderate drop is expected as removing the tool does not eliminate implicit planning, and thus represents a lower bound of its effect.
> > >
> > > Successful trajectories consistently exhibit structured planning, largely absent in weaker cases; due to space constraints, we provide only a representative example and will add systematic trajectory-level analysis in the revision.
> > >
> > > We further note that planning failures include not only missing planning, but also incorrect or inconsistent plans (e.g., flawed decomposition, dependency ordering, or blind editing in Appendix I), leading to ineffective or irrational execution. Overall, the evidence is consistent across correlation, behavior, and controlled intervention.

---

### Official Review · Reviewer_xVnq · 2026-03-18

**Soundness:** 3
**Presentation:** 3
**Significance:** 3
**Originality:** 3
**Overall Recommendation:** 4
**Confidence:** 4

**Summary:**

This paper introduce NL2Repo-Bench, a benchmark designed to evaluate the long-horizon repository generation from scratch: given only a single natural-language requirements document including structured specification (project description, supports, and API
usage guide), agents should design the architecture, manage dependencies, and produce a fully installable Python library.

The benchmark is constructed through a rigorous pipeline of repository selection, project documentation, environment building, and iterative refinement which requires significant human expertise. To ensure an equitable comparison between agents, the authors provide a reverse-engineered, quality-assured task corpus alongside a standardized evaluation image designed to isolate and eliminate external environment variables.

Experiments across SOTA open- and closed-source models reveal that long-horizon repository generation remains a significant challenge. Even the most capable agents achieve an average pass rate of only 40%, rarely completing an entire repository correctly. The analysis identifies systematic failure modes inherent to long-horizon tasks, such as premature termination, a loss of global coherence, fragile cross-file dependencies, and insufficient planning across hundreds of interaction steps.

**Compliance With Llm Reviewing Policy:**

Affirmed.

**Key Questions For Authors:**

Missing highly relevant related work: "Automatically Benchmarking LLM Code Agents through Agent-Driven Annotation and Evaluation"

**Limitations:**

yes

**Strengths And Weaknesses:**

Strength
This paper proposes to shift from evaluating localized code completion to assessing long-horizon, autonomous repository construction. By requiring agents to build from an empty workspace using only a natural language requirements document, NL2Repo-Bench provides a more authentic simulation of real-world software engineering than existing "scaffolded" datasets. Its methodology is technically rigorous, employing an AST-assisted pipeline to ensure comprehensive specifications and a Docker-based execution environment for objective, binary validation via original test suites. The evaluation results effectively expose the limitations of current state-of-the-art models, proving that even those with massive context windows struggle with global architectural coherence and cross-file dependency management.

Weakness
The benchmark is constructed from existing repositories, which introduces the risk of data contamination and may compromise the fairness of the evaluation. The benchmark is currently limited to the Python ecosystem, its findings may not fully generalize to languages with different dependency paradigms or compilation requirements. Expanding the evaluation to include additional state-of-the-art agents, such as Claude Code or Codex, would further bolster the empirical soundness of the results.

---

> ### Author Rebuttal · Authors · 2026-03-29
>
> We sincerely thank reviewer xVnq for the detailed evaluation of our work and for recognizing its contributions in different aspects. We also cherish the constructive suggestions and comments for improvement. Below we provide point-by-point responses to all the concerns raised:
>
> # Weakness1
> (Data  contamination risk)
>
> We thank the reviewer for raising the important concern regarding potential data contamination. We address this issue from three perspectives:
>
> (1) Contamination is inherent to repository-based benchmarks.
>
> We acknowledge the risk when using real-world repositories; however, this is a widely recognized limitation shared by prior benchmarks (e.g., SWE-bench and related series). Any benchmark built on open-source code may overlap with pretraining data. NL2Repo follows this standard paradigm to maintain comparability and ecological validity. Previous research also states that data contamination/leakage is a potential problem for coding-agent benchmarks[1].
>
> (2) Empirical results suggest contamination is not the main driver.
>
> Even if partial overlap exists, memorization alone cannot explain performance:
>
> Strong models achieve only ~40% success rate.
> Tasks involve extremely long contexts (repo + instructions).
> Solutions require cross-file reasoning and coordination beyond recalling isolated snippets.
>
> (3) Temporal dynamics further mitigate the risk.
>
> Selected repositories are actively maintained and may evolve beyond pretraining cutoffs. Tasks often depend on repository states that do not align with any single historical snapshot, reducing reliance on memorized content.
>
> # Weakness2
> (Limited to Python ecosystem)
>
> We thank the reviewer for this insightful comment. We agree that extending NL2Repo beyond the Python ecosystem is an important direction for improving generality, and we plan to explore multi-language settings in future work.
>
> That said, we would like to emphasize that NL2Repo evaluates a combination of language-specific skills and more general capabilities, where the latter are not tied to any particular programming language. In particular, generating a complete repository requires long-horizon planning, cross-file and cross-module consistency, and the ability to coordinate interdependent components—challenges that arise across different languages despite variations in dependency or compilation paradigms.
>
> # Weakness3
> (Additional agent framework)
>
> We agree that evaluating across multiple SOTA agent frameworks is important for strengthening the empirical soundness of the benchmark. In fact, this aspect is already incorporated in our current evaluation. While the majority of models are tested under the OpenHands-CodeAct framework, we additionally evaluate a subset of representative models using alternative agent systems, including ClaudeCode and Cursor (CLI). These results are reported as part of our main results (Sections 4.1–4.2), rather than as auxiliary experiments.
>
> Importantly, our results suggest that performance is largely consistent across different agent frameworks. For example, Claude-4.5-sonnet achieves highly similar scores across three distinct agent setups (with differences within 1 point), despite substantial differences in their tool interfaces and execution paradigms. In contrast, the performance gap between different models is significantly larger.
>
> This indicates that the observed performance differences are primarily driven by model capability rather than the choice of agent framework, providing initial evidence that our conclusions are robust across multiple widely-used coding agents.
> # Question1
>
> We thank the reviewer for pointing out this highly relevant work. We will incorporate it into our related work section of the camera-ready version paper.
>
> This paper proposes PRDBench, an agent-driven pipeline for constructing project-level benchmarks with PRD-based specifications, along with a specialized fine-tuned judge (PRDJudge) to improve evaluation reliability. We find this direction highly related and complementary to our work, particularly in moving beyond traditional unit-test-only evaluation.
>
> Compared to PRDBench, NL2Repo-Bench focuses on a different aspect of repository-level code generation. While PRDBench emphasizes scalable benchmark construction via agent-generated tasks and PRD-based evaluation, NL2Repo is grounded in real-world repositories and evaluates agents’ ability to perform cross-file reasoning and implementation under realistic development settings.
>
> We believe these two lines of work are complementary: PRDBench advances scalable and flexible evaluation methodologies, while NL2Repo-Bench provides a realistic benchmark grounded in real-world repository structures and development workflows.
>
> **References**
>
> [1]Zhou X, Weyssow M, Widyasari R, et al. Lessleak-bench: A first investigation of data leakage in llms across 83 software engineering benchmarks[J]. arXiv preprint arXiv:2502.06215, 2025.

---

> > ### Author Rebuttal · Reviewer_xVnq · 2026-04-04
> >
> > I thank the authors for the response. It addresses most of my concerns. I will keep my ratings. Good luck.

---

### Decision · Program_Chairs · 2026-04-30

**Decision:**

Accept (regular)

**Comment:**

This paper introduces NL2Repo-Bench which is an evaluation framework to test the ability of LLM agents to autonomously build fully functional Python repos from scratch only with natural language specifications. The reviewers agreed the contribution of this paper for pushing the boundaries of code generation benchmarks beyond localized infilling. Reviewers also liked the robust, docker-based evaluation methodology relying on objective upstream test suites rather than subjective LLM-as-a-judge.

During the rebuttal, the primary concerns raised included (i) the inherent risks of data contamination when utilizing public GitHub repositories, (ii) the benchmark's current limitation to the Python ecosystem, (iii) the causal validity of the paper's claims about agent planning behaviors. The authors addressed these issues by providing additional multi-seed stability metrics, token consumption statistics and a targeted ablation. While a few minor comments remain (e.g., the impossibility of entirely eliminating memorization effects), the rebuttal significantly strengthened the paper.

In conclusion, reviewers agreed that the proposed framework provides a highly valuable, technically sound simulation of real-world software engineering. Therefore, I recommend this paper for acceptance.